# Learning Long-Range Representations with Equivariant Messages

## Abstract

Machine learning interatomic potentials trained on first-principles reference data are quickly becoming indispensable for computational physics, biology, and chemistry. Equivariant message-passing neural networks, including transformers, achieve state-of-the-art accuracy but rely on cutoff-based graphs, limiting their ability to capture long-range effects such as electrostatics or dispersion, as well as electron delocalization. While long-range correction schemes based on inverse power laws of interatomic distances have been proposed, they are unable to communicate higher-order geometric information and are thus limited in applicability and transferability. To address this shortcoming, we propose the use of equivariant, rather than scalar, charges for long-range interactions, and design a graph neural network architecture, Lorem, around this long-range message passing mechanism. We consider several datasets specifically designed to highlight non-local physical effects, and compare short-range message passing with different receptive fields to invariant and equivariant long-range message passing. Even though most approaches work for careful dataset-specific choices of their hyperparameters, Lorem works consistently without adjustments, with excellent benchmark performance.

## 1 Introduction

Machine learning interatomic potentials (MLIPs) for atomistic simulations are trained on quantum-mechanical simulations to predict energies and forces of new atomic structures. Most MLIPs assume locality: The energy of each atom depends only on neighbors within a cutoff radius. This leads to linear scaling with respect to the number of atoms but fails for systems with long-range interactions, such as electrostatics, dispersion, or electron delocalization. (Behler & Parrinello; Ko et al.; Tkatchenko & Scheffler; Grisafi & Ceriotti; Unke et al., b; Huguenin-Dumittan et al.) Several approaches aim to overcome this limitation. Message-passing graph neural networks (MPNNs, Gilmer et al.) overcome locality by iteratively exchanging information between neighboring atoms, but are still constrained by the number of message-passing steps, graph connectivity, and reduced information flow with increasing number of iterations (Cai & Wang; Alon & Yahav; Nigam et al.).

An alternative approach is to use physics-inspired corrections to the total energy, written as an inverse power law of interatomic distances, $1/r^p$, with $p = 1$ for charge–charge, $p = 3$ for dipole–dipole, and $p = 6$ for dispersion. Some models predict partial atomic charges, either using explicit charge labels and equilibration schemes, or learning them implicitly from energies and forces to include electrostatic terms (Unke et al., a; Ko et al.; Fedik et al.; Pellegrini et al.; Maruf et al.; Cheng, b). Physical long-range interactions can also serve as building block: For instance, Kosmala et al. propose Ewald message passing, i.e., the use of electrostatic interactions for message passing, and Grisafi & Ceriotti propose the long-distance equivariant (LODE) framework that uses inverse power law interactions to compute equivariant features.

In this work, we aim to combine the strengths of physical $1/r^p$ interactions, computational efficiency and physically meaningful asymptotic behavior, with the ability of equivariant message passing to communicate higher-order geometric information. Extending the idea of Ewald message passing, we treat charges as equivariant objects and use inverse power law potentials as a mechanism for long-range equivariant communication. Based on this mechanism, we design Lorem, an MLIP architecture that combines short- and long-range message passing. Lorem offers consistently high

accuracy across a series of long-range benchmark tasks, outperforming other short- and long-range message passing models.

Our contributions are:

- We introduce an equivariant, global, long-range message passing mechanism that leverages existing methods from computational physics for $O(N \log N)$ scaling in periodic systems,

- We design a novel MLIP architecture, LOREM, around this mechanism,

- We conduct a series of experiments to probe the limits of equivariant short-range message passing to model long-range physics.

## 2 BACKGROUND

**Machine learning interatomic potentials**   Under the Born & Oppenheimer approximation, which decouples the nuclear and the electronic degrees of freedom, the atoms in a molecule or material move on a potential energy surface (PES) $E = E\left(\{\,(\boldsymbol{r}_i, Z_i)\,:\,i = 1...N\,\}\right)$ where $\boldsymbol{r}_i$ and $Z_i$ are positions and atomic numbers for the $N$ atoms; in a periodic system, i.e., crystals or liquids, the arrangement of $N$ atoms is repeated periodically in space, described by a cell matrix $\boldsymbol{C}$. The forces, which drive the dynamics of the atoms, are defined as the derivatives $\boldsymbol{F}_i = -\nabla_{\boldsymbol{r}_i} E$. Traditionally, this PES has been approximated by physics-inspired analytical expressions, called force fields, that are parametrized manually or through global optimization. In the past decades, in tandem with the increasing availability of large datasets of quantum mechanical reference data, MLIPs have emerged as a less computationally efficient, but more accurate, data-driven alternative.

**Atomistic graph neural networks**   Most MLIPs can be seen as graph neural networks (GNNs, Battaglia et al.) acting on a description of an arrangement of atoms in space as a geometric graph $\mathcal{G} = (\mathcal{E}, \mathcal{V})$ with edges $\mathcal{E}$ corresponding to interatomic relative-position vectors $\boldsymbol{r}_{ij}$ and nodes (or vertices) $\mathcal{V}$ corresponding to atoms. Edges are drawn between nodes that lie within a cutoff radius $r_{\mathrm{c}}$ of each other. $\boldsymbol{r}_{ij}$ are constructed to respect periodic boundary conditions. By restricting the range of interactions to neighbors on this graph, linear scaling with the number of atoms $N$ (at constant density) can be achieved; efficient scaling with system size is required to make MLIPs practical for large-scale simulations. MLIPs typically predict $E$ as a sum over atomic contributions, $E = \sum_{i=1}^{N} E_i$; the $E_i$ are usually predicted based on the node features at each $i$.

**Invariance and equivariance**   The potential energy $E$ is invariant under translations, rotations, and permutations, i.e., relabeling, of atomic positions. MLIPs must respect these symmetries, at least approximately. Translation invariance is respected by construction in atomistic GNNs. Permutation invariance is typically ensured through commutative aggregation functions, for instance sums. Finally, rotation invariance can either be learned through data augmentation (Pozdnyakov & Ceriotti; Langer et al.), or ensured by requiring that internal features remain aware of their geometric meaning, i.e., that they are *equivariant* to rotations (Smidt). While in principle, MLIPs could be constructed from invariant features only, equivariant internal features have been found to improve accuracy and data efficiency by allowing the model to access orientation information (Thomas et al.; Batzner et al.; Batatia et al. (b)). One way to construct equivariant models is to associate features with an irreducible representation of SO(3) or O(3). We denote such objects of order $l$ as tensors $\mathbf{X}^l$.

**Long-range interactions**   The potential energy $E$ arises from the many-body Schrödinger equation, which involves only Coulomb interactions between electrons and nuclei without any range separation. Efficient approximations, such as force fields or MLIPs, typically restrict interactions to local environments, motivated by the nearsightedness principle of electronic matter (Prodan & Kohn), which states that electronic properties are insensitive to distant perturbations. In the long-range regime, interactions reduce to inverse power laws $1/r^p$ of the interatomic distance. Since in nature no fixed nearsightedness length scale exists, MLIPs must capture both local many-body quantum effects, possibly extending beyond the model's cutoff $r_{\mathrm{c}}$, and formally infinite-range electrostatic interactions. Additional complexity arises from charge distributions that depend on distant atoms and from electron wavefunctions that may delocalize over large distances; see Section 6.

## 3 RELATED WORK

**Equivariant message passing**    Bond-order potentials first introduced the idea of repeatedly updating atomic environments to extend interactions beyond $r_c$ (Tersoff; Brenner), a principle now central to modern MLIPs. In MPNNs (Gilmer et al.), atoms exchange messages over $M$ steps, so features and energies depend on neighbors within $M \cdot r_c$. Early models used invariant updates (Schütt et al., a; Xie & Grossman), later extended to equivariant ones (Gasteiger et al.; Schütt et al., b; Batatia et al., b; Batzner et al.; Frank et al., b). Recent universal MLIPs trained on big datasets (Batatia et al., a; Wood et al.; Mazitov et al.; Rhodes et al.) sometimes replace message passing with local self-attention. While effective at capturing semi-local interactions, MPNNs cannot model true long-range effects, since distant atoms without intermediates never interact, and many steps reduce expressivity (Cai & Wang; Alon & Yahav; Nigam et al.).

**Physics-based long-range models**    Many long-range models explicitly add physics-inspired terms to $E$ that capture interactions decaying more slowly with distance. A common example of such a form is pairwise additive: $E^{\text{long}} = \sum_{i<j} \phi(\boldsymbol{r}_{ij};\, q_i, q_j)$, where $q_i$ are latent per atom descriptors (e.g., partial charges or polarizabilities) predicted by the model, and $\phi$ is a physically inspired, distance-based kernel. Usually, these $q_i$ are predicted directly as scalar functions of local atomic features (Unke et al., a; Staacke et al.; Loche et al.; Kim et al.; Ji et al.; Kabylda et al.; Cheng, b). A modification of this approach is to include the $q_i$ in an equilibration scheme, thus allowing these descriptors to capture information otherwise missed due to their initial dependence on local environments (Ko et al.; Pellegrini et al.; Maruf et al.). Grisafi & Ceriotti; Huguenin-Dumittan et al. propose to use physics-inspired kernels to compute long-range features, mathematically equivalent to a multipole expansion, but find that higher-order features contain little additional information.

**Other long-range models**    Some models avoid handcrafted corrections and instead learn long-range interactions directly. Ewald message passing augments GNNs with Fourier-space invariants (Kosmala et al.), while fully connected approaches use all-to-all distances (Chmiela et al.) or global attention (Unke et al., a), though these lose efficiency or distance information. Linear-scaling attention with geometric embeddings (Frank et al., a) enables global orientation exchange but needs symmetrization. Alternatives include virtual nodes for global aggregation (Caruso et al.)  or message passing in spherical harmonics space (Frank et al., c). Despite approximations, most methods still struggle to bridge periodic and non-periodic systems. Some methods outside atomistic modeling also aim to capture long-range effects (Dwivedi et al.; Bamberger et al.; Moskalev et al.; Zhdanov et al.).

## 4 EQUIVARIANT LONG-RANGE MESSAGE PASSING

Ewald summation, as introduced in Appendix C, allows the efficient, and convergent, evaluation of a potential at each atomic position, $V_i = V(\boldsymbol{r}_i) = \sum_{j=1}^N \sum_{\boldsymbol{n} \in \mathbb{Z}^3} q_j / |\boldsymbol{r}_i - (\boldsymbol{r}_j + \boldsymbol{Cn})|^p$. From the perspective of machine learning, this computation can be seen as a basic form of message passing, with radial filter $1/r_{ij}^p$, neighbor feature $q_i$, and resulting message $V_i$; this correspondence was pointed out by Kosmala et al. and used by Grisafi & Ceriotti to compute general long-range features.

To bring it more in line with standard message passing, the operation can be carried out across an extra channel dimension, promoting a single feature $q_i$ to a vector $\boldsymbol{q}_i$. This allows to communicate more information, but in this form, this information is restricted to geometric invariants. We propose to promote $q_i$ to an *equivariant* tensor instead: $\mathbf{Q}_i^l$. This results in equivariant messages, or potentials:

$$\mathbf{V}_i^l = \sum_{j=1}^N \sum_{\boldsymbol{n} \in \mathbb{Z}^3} \frac{\mathbf{Q}_j^l}{|\boldsymbol{r}_i - (\boldsymbol{r}_j + \boldsymbol{Cn})|^p} \,. \tag{1}$$

To see that the result is equivariant, recall that adding two equivariant objects yields another equivariant object, and that multiplication with a prefactor, provided that the factor is shared across all entries for a given $l$, also retains equivariance. It is then easy to see that multiplying $\mathbf{Q}_i^l$ with a factor $1/r_{ij}^p$ yields an equivariant, and that the sum in Eq. (1) also yields an equivariant, since all summands are equivariant objects. We argue and numerically confirm in Appendix A that a compensating background charge correction preserves equivariance.

Overall, this approach allows the global aggregation of equivariant messages in a way that is amenable to efficient implementation, scaling $O(N \log N)$ in periodic systems and – at least in principle, see Appendix C – $O(N)$ in finite systems. The method also naturally accommodates the asymptotic power-law decay intrinsic to most long-range effects in physical systems.

## 5 LOREM

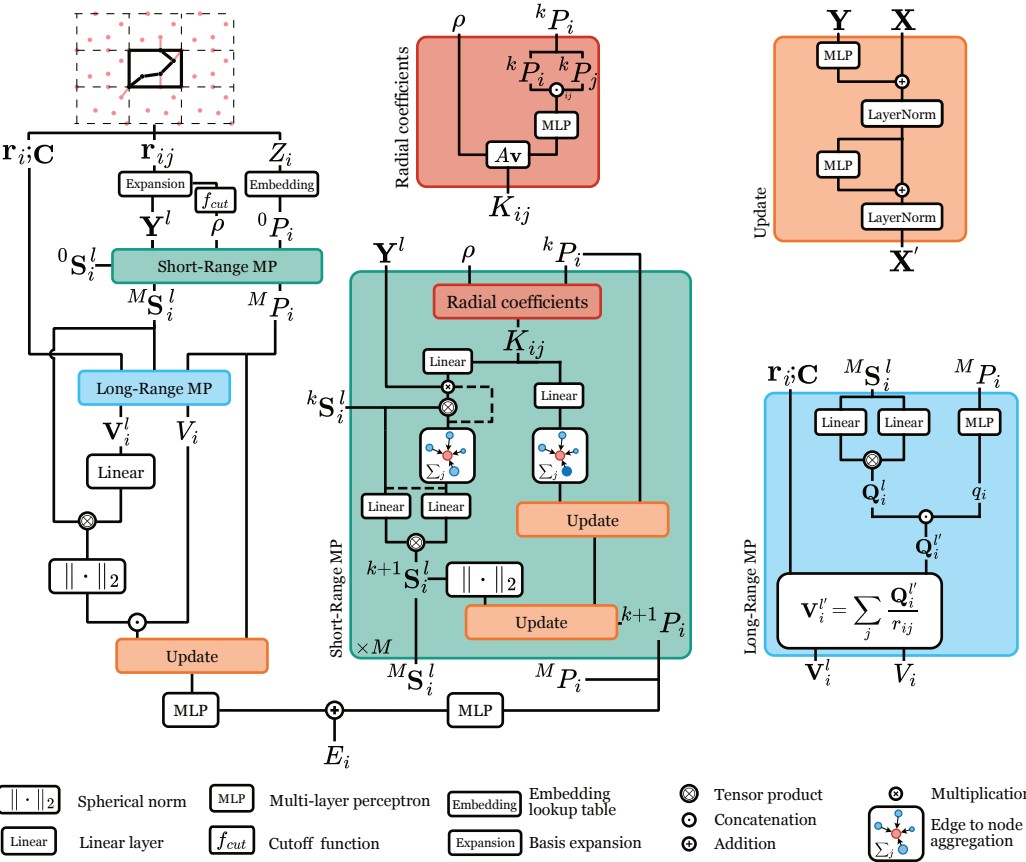

Figure 1: Sketch of the LOREM architecture.

LOREM follows the blueprint of equivariant MPNNs, making a few modifications: Scalar and spherical features are handled separately, following Frank et al. (c;b), and mixed using a variant of the power spectrum (Bartók et al.) and a residual update block (He et al.).

**Overview**   LOREM, illustrated in Fig. 1 maps a point cloud of $N$ atomic positions $\{\, r_i \,\}$, potentially contained in a periodic unit cell $C$, and labeled with chemical species $\{\, Z_i \,\}$ to atomic energies $\{\, E_i \,\}$. Internally, the point cloud is processed as a graph, with connectivity between nodes, the atoms, defined by a cutoff radius $r_c$. Accordingly, edges correspond to vectors $r_{ij} = r_j - r_i$. Node features are updated through either **short-range message passing** or **long-range message passing**. Spherical information is used to update scalar features by computing their **spherical norm** and passing it through an **update** block. Updates of scalar features are followed by a residual prediction of atomic energy contributions. Selected blocks are discussed below; the remaining in Appendix E.

**Short-range message passing**   Distances $r_{ij}$ are expanded in Bernstein polynomials multiplied with $f_{cut}$, the cosine cutoff function, yielding $\rho_{ij}$. Orientations $r_{ij}$ are expanded in spherical harmonics $Y^l_{ij}$. Initial scalar features are a learned embedding of chemical species. At each short-range message

passing step,[1] edge features $\boldsymbol{K}_{ij}$ are obtained by concatenating ${}^k\boldsymbol{P}_i$ and ${}^k\boldsymbol{P}_j$. They are the input for a learned linear transformation of the radial basis; the result is summed to update node features. Linearly transformed edge features are also used as prefactors for $\mathbf{Y}_{ij}^l$, which participate in a tensor product with ${}^k\mathbf{S}_j^l$; the results are aggregated into a spherical message, which updates the spherical features via a tensor product, resulting in ${}^{k+1}\mathbf{S}_i^l$. The spherical norm of the updated spherical node features is then used to further update the scalar features, finally yielding ${}^{k+1}\boldsymbol{P}_i$. This process is repeated $M$ times (the total number of short-range message passing steps).

**Long-range message passing**   To minimize computational cost, which is proportional to the number of channels over which Ewald summation is carried out, node features are transformed into low-dimensional charges: The scalar features ${}^M\boldsymbol{P}_i$ are transformed into a single charge $q_i$, and spherical features ${}^M\mathbf{S}_i^l$ are likewise, using a linear transformation followed by a self tensor product, transformed into $\mathbf{Q}_i^l$, typically with a reduced maximum spherical harmonics degree and singular feature dimension. Unless otherwise noted, we use $l_{\max} = 2$, which yields a total of 10 charge channels.[2]  After Ewald summation, which is carried out in parallel across charge channels, the potentials are split into the purely scalar $V_i$ and the spherical $\mathbf{V}_i^l$. The spherical potentials are combined with spherical node features through a tensor product; the spherical norm of the result is concatenated with the scalar potential to update scalar representations. We find that using only $p{=}1$, i.e., Coulomb interaction, rather than a set of different exponents, is sufficient in practice.

## 6  Experiments

We perform experiments on a number of existing benchmark tasks designed to probe the ability of MLIPs to model long-range interactions, comparing Lorem to short and long-range MLIPs. The experiments are divided into two parts: In Section 6.1, we compare Lorem with other models using standard settings. We find that Lorem performs well, but also observe that most tasks can be solved by models that do not consider long-range interactions at all – the effective interaction range of typical message passing models is sufficient. However, predictions break down beyond this interaction range. In Section 6.2, we probe this limit of short-range message passing. We find that while careful consideration of hyperparameters is required to resolve long-range interactions with short-range models, Lorem can solve these benchmarks without adaptation.

**Models**   We compare Lorem with a number of purely short-ranged MLIPs: Mace and Pet, as well as the recently introduced Cace-Les model that combines short-range message passing with a scalar long-range part, and 4G-NN, which includes a physics-based long-range energy contribution and charge equilibration. Full details on model descriptions and training can be found in Appendix F.

The datasets and associated benchmark tasks used in our experiments are briefly introduced below; additional details are given in Appendix B. An overview of validation or test set (where available) metrics for all datasets is given in Table 1. Lorem performs very well across datasets, and is competitive with, or more accurate than, other models.

**MgO surface**   This benchmark task is the first in a series designed by Ko et al. to highlight the need for long-range information in MLIPs. Illustrated in Fig. 2A, it consists of a magnesium oxide (MgO) surface on which a gold ($Au_2$) dimer is placed. Depending on the presence of an aluminum (Al) dopant deep inside the surface, the lowest-energy position of the gold dimer is either in the 'wetting' (flat), or 'non-wetting' (standing up), position. The benchmark consists of two parts: Correctly identifying the ordering between wetting and non-wetting geometries in the doped and undoped case, and reproducing the energy-distance curve for the non-wetting geometry, in particular the local minimum corresponding to the equilibrium distance.

---

[1]In the initial message passing step, no spherical node features are available, and therefore the tensor product with $\mathbf{Y}_{ij}^l$ is omitted and initial spherical node features are obtained via a self tensor product (dotted lines).

[2]Two with $l{=}0$: one from the scalar features and one from the scalar component of the spherical features.

[3]4G-NN training requires DFT-computed charges, which are not available for the biodimers and cumulene.

Table 1: Root mean squared errors for energy $E$ and forces $\boldsymbol{F}$ for datasets and models used in Section 6.1. Where available, a held-out test set was used; for small datasets, the validation set was used instead and indicated in the table. The lowest error is indicated in bold. The second line after each model name indicates the number of short-range (SR) message-passing steps and whether some form of long-range (LR) interactions are included in the model.[3]

|  | Dataset | LOREM $1\times$SR + LR | CACE-LES $1\times$SR + LR | MACE $2\times$SR | PET $2\times$ SR | 4G-NN $1\times$SR + LR |
|---|---|---|---|---|---|---|
| MgO surface | $E$ (meV/at) | **0.064** | 0.071 | 0.376 | 0.210 | 0.219 |
| (Validation set) | $\boldsymbol{F}$ (meV/Å) | **4.076** | 7.913 | 5.971 | 6.261 | 66.000 |
| NaCl cluster | $E$ (meV/at) | **0.112** | 0.210 | 1.681 | 1.517 | 0.481 |
| (Validation set) | $\boldsymbol{F}$ (meV/Å) | **1.155** | 9.784 | 40.219 | 42.438 | 32.780 |
| Biodimers | $E$ (meV/at) | **0.222** | 2.259 | 7.793 | 6.758 | – |
|  | $\boldsymbol{F}$ (meV/Å) | **1.646** | 3.163 | 16.150 | 16.470 | – |
| Cumulene | $E$ (meV/at) | 3.309 | 17.803 | 12.592 | **3.205** | – |
|  | $\boldsymbol{F}$ (meV/Å) | 50.084 | 147.616 | 104.318 | **46.905** | – |

**NaCl cluster**   Similar to the presence of a dopant modifying the potential-energy surface for the gold dimer in the MgO surface task, this benchmark by Ko et al. relies on the presence or absence of a sodium atom at one end of a charged sodium chloride (NaCl) cluster changing the behavior of a sodium atom at the opposite end: Since one atom is removed while the charge remains constant, the charge must redistribute over the remaining atoms. The benchmark task consists of reproducing the location of the local minimum and the energy profile when moving the sodium atom farthest from the removed one, indicated in Fig. 2C.

**Cumulene**   The cumulene benchmark task was proposed by Unke et al. (b) as an example of a long-range problem that is due to a non-local effect of the electronic structure of a molecule. This molecule consists of a chain of nine carbon atoms, with a pair of hydrogen atoms, the rotors, at the opposite ends. The orientation of the rotors determines the shape of the atomic orbitals for each carbon, which propagates along the chain to the other end; in the absence of bending or stretching, the energy is therefore fully determined by the relative orientation of the rotors. The task is to recover the energy profile of this rotation for an idealized fully extended geometry, illustrated in Fig. 3.

**Biodimers**   This benchmark dataset by Huguenin-Dumittan et al. consists of pairs of relaxed organic molecules placed at distances of $4\,\text{Å}$ to $15\,\text{Å}$ from each other, illustrated in Fig. 4A. Depending on the chemical nature of the molecules, interactions between the molecules take different asymptotic power law forms, ranging from charge-charge $1/r$ interactions to apolar-apolar $1/r^6$ interactions. Here, we simply compare energy and force prediction errors on test sets stratified by the dominating power-law interaction.

**$S_N2$ reactions**   Finally, the $S_N2$ reactions benchmark was introduced by Frank et al. (a) to probe the ability of MLIPs to model the long-range interactions required to mediate gas-phase chemical reactions. It consists of the nucleophilic substitution of methyl halides by another halide ion: $X^- + H_3C{-}Y \rightarrow X{-}CH_3 + Y^-$ where X,Y = F, Cl, Br, or I. The benchmark task in this case is predicting the energy along the reaction coordinate, i.e., correctly modeling the potential energy profile as the two reactants approach one another, react, and separate again.

### 6.1 STANDARD SETTINGS

We compare LOREM to other models using their standard settings (see Appendix F).

**MgO surface**   The results for this experiment can be seen in Fig. 2B: Despite being designed to require long-range interactions, this benchmark task can be solved by both purely short-range and long-range message-passing models. In this case, the success of short-range message passing is due to the small size of this benchmark system: With an effective cutoff radius above $6\,\text{Å}$, centrally

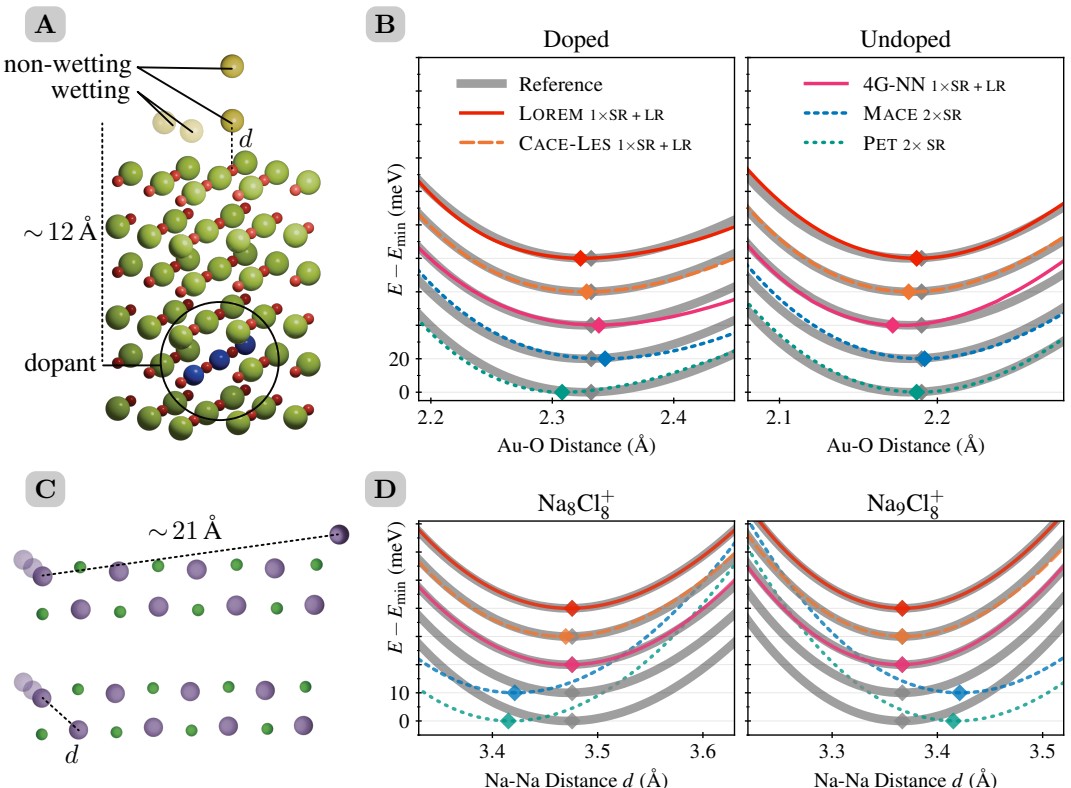

Figure 2: (**A**) $Au_2$ dimer on MgO surface, showing both wetting and non-wetting geometries, as well as the Al dopant. (**B**) Energy over distance $d$ for the non-wetting geometry for the doped and undoped surface. The minima are indicated with a diamond symbol; the reference energy curve is drawn in grey. Offsets are added to distinguish the curves and the value at the minimum is subtracted. (**C**) $Na_9Cl_8^+$ (top) and $Na_8Cl_8^+$ (bottom) cluster, the moving atom is marked with transparent copies of itself, and the distance of interest is labeled with $d$. (**D**) Energy over distance for both clusters.

located atoms can 'see' the full system and hence, MACE and PET can solve this task. All models also resolve the orientation preference for the $Au_2$ dimer between the doped and undoped surfaces.

**NaCl cluster**    The results of this experiment, seen in Fig. 2D, are drastically different from the previous one: Here, only models with a long-range component are able to resolve the difference between $Na_9Cl_8^+$ and $Na_8Cl_8^+$. All such models show excellent agreement with reference values. The failure of short-range message-passing is due to the larger system size compared to the MgO surface: Here, effective cutoff radii exceeding $10.5\,\text{Å}$ are required to solve the benchmark task.

**Cumulene**    Resolving the cumulene energy profile, seen in Fig. 3, requires simultaneous knowledge of the orientation of both rotors at opposite ends of the molecule. This can be achieved in two ways: Through equivariant short-range message passing (MACE, PET), provided that the chain is not too long (see Section 6.2), or through *equivariant* long-range message passing (LOREM). For this reason, CACE-LES cannot solve this benchmark: Scalar charges are not sufficiently expressive to communicate relative orientation. All equivariant models that are able to access this information can solve this benchmark, achieving good agreement with the reference data. PET resolves the dihedral angle as well, but requires long training and an adaptation in the number of transformer layers to succeed at this benchmark (see Appendix F). We note that the sharp cusp at $180°$ is an artifact of the underlying reference method; it is a desirable behavior of MLIPs to smoothen it out.

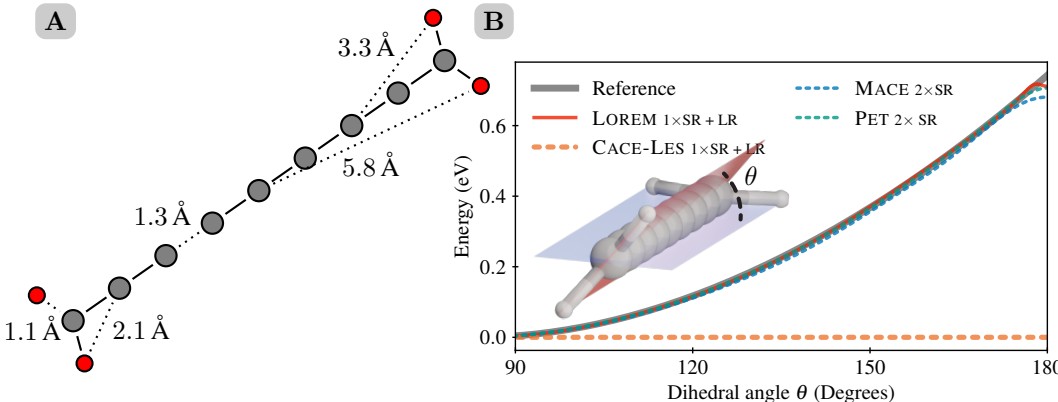

Figure 3: (**A**) Illustration of cumulene laid flat, indicating relevant distances between atoms. (**B**) Energy profile over a 90° rotation of one rotor. The minimum value of each curve is subtracted before plotting. The inset shows a 3D representation cumulene, defining the dihedral angle $\theta$.

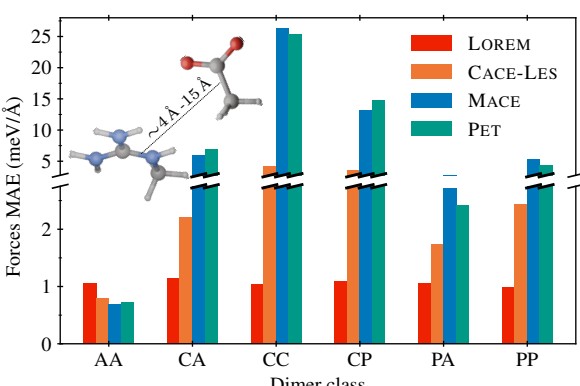

Figure 4: Mean absolute error on forces for different models on the different dimer classes: Apolar-apolar (AA), charge-apolar (CA), charge-charge (CC), charge-polar (CP), polar-apolar (PA), polar-polar (PP). A charge-charge pair is shown as inset.

**Biodimers** Since the pairs of molecules in the biodimers benchmark are placed at separations up to $15\,\text{Å}$, much beyond the cutoff used for graph construction, high accuracy requires a long-range component. This is confirmed by the results of this experiment, seen in Fig. 4: The long range LOREM and CACE-LES models yield lower error than MACE and PET. Predictive error varies between dimer classes, i.e., the expected type of inverse power law interaction, for all models, with the exception for LOREM, which yields consistent, and in all but one class the highest, accuracy. This may be due to the addition of a nonlinearity after long-range message passing, which allows the model to correct the mismatch between the power law used for message passing, $1/r$, and the different exponents in the dimer classes.

## 6.2 LIMITS OF SHORT-RANGE MESSAGE PASSING

In the previous experiments, we observed that the performance of models with short-range message passing strongly depends on the match between the effective interaction cutoff and the problem to be solved. The case of biodimers, demonstrates clearly that message passing cannot resolve interactions where no intermediate atoms are present. To study these cases, we perform additional experiments using LOREM, with and without long-range message passing, and with different cutoffs.

**Cumulene with different cutoffs** We probe the dependence of the performance of message-passing models on their hyperparameters by training a set of LOREM models with and without long-range message passing at different cutoffs ($2.5\,\text{Å}$, $3\,\text{Å}$ and $3.5\,\text{Å}$) and numbers of short-range message passing steps (1, 2, 3 and 4). Small cutoffs are chosen to simulate the longer chain lengths of real biomolecular systems. The results are shown in Table 2 of Appendix D: In all cases, models that include long-range message passing are able to resolve the angle. Short-range models, on the other hand, can only resolve the angle in certain combinations of hyperparameters: One message passing step is never sufficient. For two, a minimum cutoff of $3.5\,\text{Å}$ is required. For three message passing steps, $3.0\,\text{Å}$ is required. Therefore, an effective cutoff substantially larger than the $5.8\,\text{Å}$ distance

between the central carbon atom in the chain and the hydrogen rotors is needed. This is because MPNNs rely on the input graph structure (see Fig. 3A) for information flow: Below $r_c = 2.6\,\text{Å}$, connections in the graph only extend to nearest neighbors, with the exception of the rotors and the second-to-last carbon atoms. Consequently, at least four message passing steps are required. This example illustrates that short-range message passing is difficult to apply to this class of problems: Parameters have to be adapted to the chain length, with careful consideration of graph structure. In contrast, long-range message passing is robust, requiring no change in parameters to solve this task.

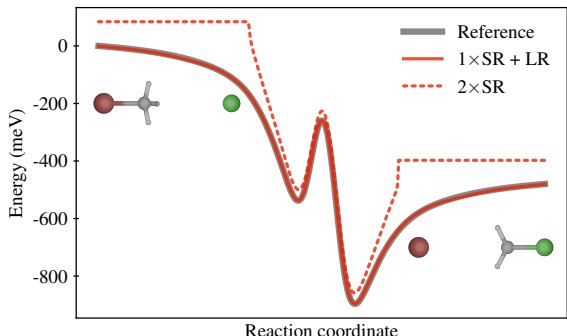

Figure 5: Energy over the reaction coordinate for the nucleophilic substitution reaction $Cl^- + H_3C-Br \rightarrow Cl-CH_3 + Br^-$; snapshots are shown as insets.

**$S_N2$ reactions** Solving the benchmark task for the $S_N2$ reactions dataset requires long-range interactions, since it involves modeling intra-molecular interactions over distances exceeding the typical cutoffs used for MLIPs. This is confirmed by Fig. 5, which compares the performance of LOREM with and without long-range message passing. The former can reproduce the energy over the course of the reaction with excellent accuracy, including the tails as the reactants approach and separate. The latter, which does not include long-range interactions, cannot account for the tails and consequently predicts a constant once the molecules separate more than $5\,\text{Å}$.

# 7 DISCUSSION

We introduced a simple yet effective equivariant long-range message passing scheme: Latent equivariant charges are predicted from local node features, and well-established techniques for evaluating inverse power-law potentials are used to efficiently compute long-range messages. Building on this, we developed LOREM, a MLIP architecture that achieves consistently strong performance across benchmarks that require accurate modeling of long-range interactions.

By construction, our model assumes that interactions decay asymptotically with distance. While it is therefore not suited for learning truly global representations that have no notion of locality, this limitation is largely theoretical for physical systems: Electrostatics dominates most long-range behavior in realistic systems, and even other effects typically do not extend over arbitrary distances. A more practical limitation arises in non-periodic systems, where the cost of long-range message evaluation scales as $O(N^2)$. Although this is acceptable for small molecules, this may become a bottleneck for larger systems. Linear scaling can be achieved using fast multipole methods (Greengard & Rokhlin; Andy L Jones) or alternative methods like Euclidean Fast Attention (Frank et al., a).

We also investigated the capabilities of purely short-range message passing and found that, in many cases, it performs very well—even on datasets explicitly designed to require long-range interactions or charge equilibration. However, its success depends on matching the cutoff radius and number of message passing steps to the specific task, a process that can be both tedious and error-prone. More fundamentally, short-range methods cannot resolve interactions between distant atoms without intermediaries. Our augmentation with long-range message passing overcomes these limitations.

Finally, our results underscore a broader issue: the lack of challenging long-range benchmarks. Many current datasets are based on simplified, small-scale systems and fail to capture the complexity of real-world applications where long-range interactions are essential. Addressing this gap is a key direction for future work.

REPRODUCIBILITY STATEMENT

Code, data, configuration files, and trained models are available at doi:10.5281/zenodo.17194573. Scripts include data pre-processing, model training, model evaluation, and the creation of figures and tables in this work. Hyperparameters are also described in Appendix F.

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

## A    INVARIANCE OF LOREM

As explained in Section 4, the potential at each atom, $\mathbf{V}_i^l$ is equivariant because the long-range message passing step consists of equivariant operations: Addition, and multiplication with a factor shared per $l$. In practical implementations of Ewald summation in periodic systems, there is one extra step: To prevent divergences, the total charge must be zero, which can be done either by simply subtracting the sum of the total charge from each charge, or equivalently, by subtracting an analytical correction from the potentials. Both approaches are equivalent, and since summation is equivariant, also do not break invariance. For this reason, the total procedure retains equivariance.

To numerically confirm these considerations, we compute the mean absolute error of energy predictions over a degree $L = 3$ Lebedev grid of rotations, plus inversions, with respect to the unrotated case, for the first 5 structures of the MgO surface validation set. The errors are in line with precision expectations: For single precision, $1.031 \cdot 10^{-6}$ meV and for double precision $5.913 \cdot 10^{-15}$ meV.

## B    DATASET CONSTRUCTION

**MgO surface**    The MgO surface dataset from Ko et al. is used without modification. A representative snapshot of the unit cell is shown in Fig. 2A. The dataset is built from four configuration types, each derived from a distinct initial structure:

1. A pure MgO surface with the $Au_2$ dimer oriented perpendicular to the surface,

2. A pure MgO surface with the $Au_2$ dimer oriented parallel to the surface,

3. An Al-doped MgO surface with the $Au_2$ dimer perpendicular to the surface, and

4. An Al-doped MgO surface with the $Au_2$ dimer parallel to the surface.

Each of these initial configurations was first geometry optimized. For the two perpendicular, 'non-wetting', cases (1 and 3), the distance between the lower Au atom and the O atom directly beneath it was systematically varied, displacing the $Au_2$ dimer as a whole. From these distance-dependent samples, a subset was randomly selected. To introduce structural diversity, Gaussian noise was applied to each configuration: a standard deviation of $0.02$ Å for atoms in the MgO substrate and $0.1$ Å for the gold cluster. After perturbation, 1250 structures were selected from each of the four configuration types, yielding a total of 5000 samples. For these, energies and forces were computed using the FHI-aims code (Abbott et al.) and the Perdew–Burke–Ernzerhof (PBE) functional (Perdew et al.). A random 90/10 train–validation split was used, as reported in Ko et al.. Additional energy–distance curves with equal spacing were constructed for the perpendicular cases and are shown in Fig. 2B.

**NaCl cluster**    The NaCl cluster dataset from Ko et al. is used without modification. First, the $Na_9Cl_8^+$ cluster was optimized in vacuum. A snapshot is shown in the top panel of Fig. 2C. From this geometry, the Na atom farthest from all other Na atoms was removed, yielding the $Na_8Cl_8^+$ cluster shown in the bottom panel of Fig. 2C. From these two structures, additional configurations were created by varying the distance between a selected pair of Na atoms along the line connecting them. In Fig. 2C, this moving atom is illustrated by transparent copies along its trajectory. Training datasets for both $Na_9Cl_8^+$ and $Na_8Cl_8^+$ were constructed by randomly sampling configurations along these trajectories. Gaussian noise with a standard deviation of $0.05$ Å was applied to the atomic coordinates, resulting in 2500 perturbed structures for each molecule and a total of 5000 instances. Energies and forces were computed using the FHI-aims code and the PBE functional. A random 90/10 train–validation split was used, as reported by Ko et al.. Additional energy–distance curves with equal spacing were constructed and are shown in Fig. 2D.

**Cumulenes**    The cumulene dataset from Unke et al. (b) is used without modification. The geometry of the linear molecule is shown in Fig. 3A. The dataset contains 4500 randomly sampled cumulene structures with nine carbon atoms, divided into training, validation, and test sets with 2000/500/2000 instances, respectively. The energy curves shown in Fig. 3B are based on a controlled subset in which only one terminal carbon atom is rotated, while the rest of the molecule remains fixed. A visualization of this rotational motion is provided in the inset of Fig. 3B.

**Biodimers** The Biodimers dataset from Huguenin-Dumittan et al.; Burns et al. (2017) is used without modification. It consists of 2291 relaxed organic sidechain–sidechain fragments, including small molecules such as ethanol, acetamide, and others. Based on molecular properties, the dataset is divided into six categories: each molecule is classified as either polar, apolar, or charged, resulting in the following dimer types—polar–polar (PP), polar–apolar (PA), polar–charged (PC), apolar–apolar (AA), apolar–charged (AC), and charged–charged (CC). A representative CC dimer is shown in Fig. 4A. The initial separation between molecules reflects their positions in protein structures. From this configuration, the intermolecular distance is incrementally increased up to $15\,\text{Å}$, resulting in a total of $29\,783$ dimer instances. Instances where the final separation exceeds the initial distance by more than $4\,\text{Å}$ ($13\,743$ samples) are designated as the test set. The remaining $16\,040$ instances form the training set. Energies and forces were computed using the FHI-aims code and the HSE06 hybrid functional. For each of the six dimer types, a random 80/20 train–validation split was applied. The resulting subsets were then merged into a single training set and a single validation set.

**$S_N2$ reactions** The $S_N2$ reactions dataset from Unke & Meuwly; Frank et al. (a) is used without modification. It contains molecular structures in vacuum that model nucleophilic substitution ($S_N2$) reactions. The dataset includes molecules of the types $XCH_3Y^-$, $CH_3X$, $HX$, $CHX$, $CH_2X$, $XY$, $X$, and $Y$, for all possible combinations of $X, Y \in \{F, Cl, Br, I\}$. A representative reaction coordinate with corresponding snapshots is shown in Fig. 5. Additional species such as $H_2$, $CH_2$, and $CH_3$ are also included in the dataset. Training configurations were generated using ab initio molecular dynamics simulations at $5000\,\text{K}$, with a time step of $0.1\,\text{fs}$. Full computational details, including the level of theory, are provided in the original publications. The dataset is randomly split into $405\,000$ training, $5000$ validation, and $42\,708$ test samples.

## C  Ewald summation

The evaluation of inverse power law potentials has been the object of much study in computational physics and chemistry. The general task is to compute the potential $V$ and the potential energy $E$:

$$V(\boldsymbol{r}) = \sum_{j=1}^{N} \sum_{\boldsymbol{n} \in \mathbb{Z}^3} \frac{q_j}{|\boldsymbol{r} - (\boldsymbol{r}_j + \boldsymbol{C}\boldsymbol{n})|^p} \qquad E_{\text{LR}} = \sum_i \frac{1}{2} q_i V(\boldsymbol{r}_i)\,, \qquad (2)$$

where $q_i$ are charges, or other coefficients, placed on each atomic position, and $\boldsymbol{n}$ is a vector indicating periodic replicas. For $p \leq 3$, this infinite sum converges conditionally, i.e., convergence or divergence depends on the summation order. Ewald summation was developed to tackle this problem for electrostatics ($p = 1$) and later extended to other exponents; its basic concept is to split $1/r$ into a short-ranged part, which converges fast in real space, and a long-range part, which is smooth, and therefore converges well, in reciprocal space.

A naive implementation of this method scales $O(N^2)$, which can be brought down to $O(N^{3/2})$ by choosing the cutoffs to be proportional to the size of the simulation cell. Particle–mesh Ewald (PME, P3M) algorithms reduce this to $O(N \log N)$ by interpolating charges onto a grid and employing the fast Fourier transform for the reciprocal-space part (Darden et al.; Hockney & Eastwood). While such methods are standard in force fields, implementations in popular machine learning frameworks, PyTorch (Paszke et al.) and JAX (Bradbury et al.), have only become available recently (Loche et al.). In non-periodic systems, the naive evaluation of Eq. (2) scales $O(N^2)$. However, methods like the fast multipole expansion can reduce this cost to $O(N)$ (Greengard & Rokhlin).

## D  Additional results for cumulene

In Fig. 6, we present the curves related to Table 2. All models where the dihedral information is accessible can resolve the benchmark, but vary in match to the ground truth. In particular, we note that models with two short-range message passing steps perform better than a single one. This may be due to the high degree of degeneracy in atomic environments at short cutoffs, which is alleviated by message passing.

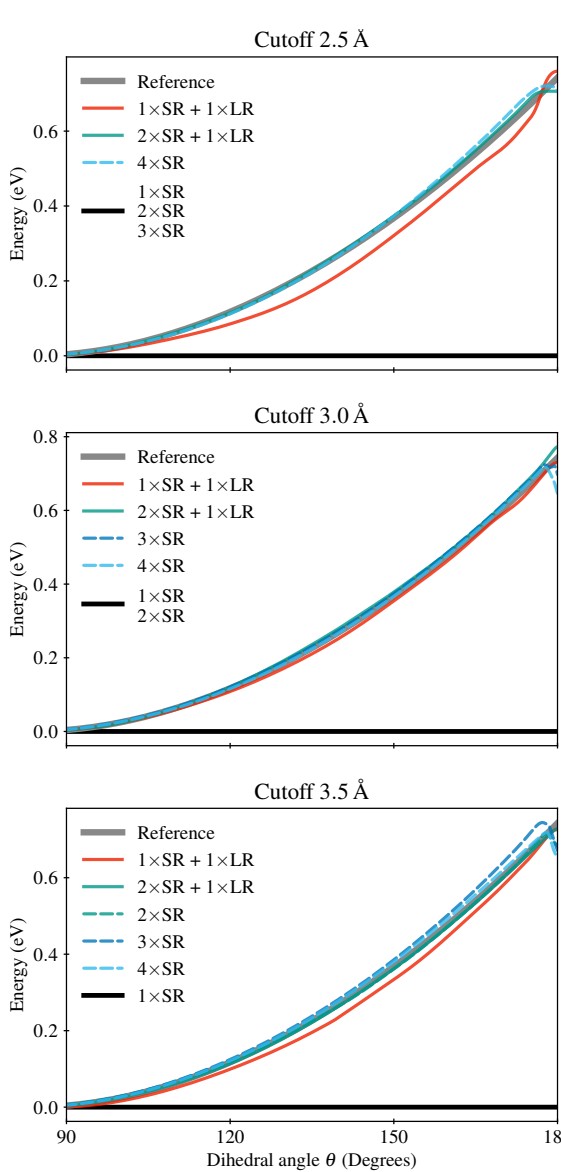

Figure 6: Rotational profile of cumulene for variations of the LOREM model with different $r_c$, different numbers of short-range message passing steps, and with and without long-range message passing. All curves that are identical to zero have been collapsed into a single one for readability.

Table 2: Ability of different LOREM models, with and without long-range message passing and with different numbers of short-range message passing steps, to solve the cumulene benchmark task. A tick ($\checkmark$) indicates yes, a cross ($\times$) indicates no.

| Cutoff | $1\times$SR + LR | $2\times$SR + LR | $1\times$SR | $2\times$SR | $3\times$SR | $4\times$SR |
|---|---|---|---|---|---|---|
| $2.5\,\text{Å}$ | $\checkmark$ | $\checkmark$ | $\times$ | $\times$ | $\times$ | $\checkmark$ |
| $3.0\,\text{Å}$ | $\checkmark$ | $\checkmark$ | $\times$ | $\times$ | $\checkmark$ | $\checkmark$ |
| $3.5\,\text{Å}$ | $\checkmark$ | $\checkmark$ | $\times$ | $\checkmark$ | $\checkmark$ | $\checkmark$ |

## E  LOREM BLOCKS DESCRIPTION

Below is the description of the blocks that were not described in the main text:

**Spherical norm**  For any set of spherical (equivariant) features, i.e., possessing an index $l$ and for each $l$ a corresponding index $m$ (see Section 2), the spherical norm is defined as:

$$\|\mathbf{S}^l\|_2 = \sqrt{(2l+1)^{1/2} \sum_{m=-l}^{l} S_{lm}^2}\,.$$

We find that the empirically determined prefactor reduces variance across $l$.

**Update block**  Updates $\boldsymbol{Y}$ to scalar features $\boldsymbol{X}$ are processed with an update block:

$$\boldsymbol{X} \leftarrow \boldsymbol{X} + \text{MLP}(\boldsymbol{Y})$$
$$\boldsymbol{X} \leftarrow \text{LayerNorm}(\boldsymbol{X})$$
$$\boldsymbol{X} \leftarrow \boldsymbol{X} + \text{MLP}(\boldsymbol{X})$$
$$\boldsymbol{X} \leftarrow \text{LayerNorm}(\boldsymbol{X})\,,$$

following a residual structure.

## F  MODEL DESCRIPTION AND TRAINING SETUP

**LOREM**  For all experiments, except the ones where variations were tested, the same LOREM model architecture (Section 5) was used: A cutoff radius of $5\,\text{Å}$, $l_{\max} = 6$ for spherical features and $l_{\max} = 2$ for the long-range message passing, 128 scalar features, 8 channels for spherical features, and 32 radial basis functions. We perform only the initial short-range message passing step. This model has $1\,021\,198$ learnable parameters. It was implemented using the `e3x` library (Unke & Maennel). Training is performed entirely in `float32` precision; while we find that reduced precision has only a minor effect on training dynamics, it can significantly alter validation and test results.

Training parameters vary slightly between datasets and were chosen by hand to minimize validation error;[4] the exact parameters can be found in Table 3. We report errors for the better out of two runs with different seeds; the presented conclusions hold for both models. In all cases, except NaCl[5], the learning rate was decayed linearly after 10 epochs, from the indicated starting value to $1 \cdot 10^{-6}$; the ADAM (Kingma & Ba) and LAMB (You et al.) optimizers were used. The loss function was a simple mean squared error, with the energy residuals normalized by number of atoms. The squared residuals were averaged over the whole batch, including over atoms and components in the case of forces, and then summed and weighted with a factor. The checkpoint with the lowest summed $R^2$ of energy and forces, evaluated on the validation set, was used for experiments. Training times are given for the entire run, not the time until the best checkpoint.

---

[4]We note that the validation error in the cumulene dataset does not readily correlate with ability to predict the dihedral curve in Fig. 3 – while all models are able to resolve the task, the agreement varies.

[5]Here, exponential learning rate decay was employed.

| Dataset | Optimizer | Initial LR | Epochs | Batch size | $E$ weight | $F$ weight | Time |
|---|---|---|---|---|---|---|---|
| MgO surface | ADAM | $4 \cdot 10^{-4}$ | 4000 | 32 | 1000 | 1.0 | 17 h |
| Biodimers | ADAM | $1 \cdot 10^{-4}$ | 4000 | 32 | 0.5 | 0.5 | 42 h |
| Cumulene | LAMB | $1 \cdot 10^{-3}$ | 2000 | 32 | 0.5 | 0.5 | 2 h |
| NaCl cluster | ADAM | $1 \cdot 10^{-3}$ | 8000 | 64 | 0.5 | 0.5 | 7 h |
| $S_N 2$ reactions | ADAM | $1 \cdot 10^{-4}$ | 500 | 32 | 0.5 | 0.5 | 17 h |

Table 3: Training settings for LOREM for different datasets. Training times are given for a single Nvidia H100 SXM5 GPU.

**MACE** The MACE MLIP (Batatia et al. (b)) is an equivariant message-passing neural network as described in Section 2. It is used with the standard setting of two message-passing steps, i.e., an effective cutoff of $10 \, \text{Å}$. Training generally used default hyperparameters, with some adjustments made on hyperparameters related to training dynamics. Specifically, the cumulene system was trained using the default energy-to-forces loss weight ratio of 1:100 and '128x0e + 128x1o + 128x2e' hidden irreps. All other systems were trained using a 1:10 ratio and '128x0e + 128x1o' hidden irreps.

All experiments, except those involving cumulenes and the NaCl cluster, used the SWA protocol, which swaps the loss weights between energy and forces at a specified epoch. This epoch was set where the energy RMSE plateaued, with the subsequent training continued until loss saturation. All models were trained with $l_{\max} = 2$, a batch size of 32, and utilized cuEquivariance acceleration.

**PET** PET (Pozdnyakov & Ceriotti) is an unconstrained transformer model, consisting of multiple edge-to-edge transformer layers within local neighborhoods followed by message passing. Similar to MACE, it is also used with two message-passing steps and an effective cutoff of $10 \, \text{Å}$. To ensure approximate rotational invariance, it is trained with data augmentation. No inference-time symmetrization is used in our experiments.

Unless otherwise specified, we used default hyperparameters (cutoff radius of $5 \, \text{Å}$ with cosine smoothing over the outermost $0.2 \, \text{Å}$, $d_{\text{PET}} = 128$, $d_{\text{head}} = 128$, $d_{\text{feedforward}} = 512$, with 8 heads per attention layer, and 2 attention layers per GNN layer), with some adjustments made related to training dynamics. Models were trained using an epoch-based scheduler, which halved the learning rate after 250 epochs. This applies to all datasets except biodimers. For biodimers, a `ReduceLROnPlateau` scheduler was used instead, reducing the learning rate by 20% if the loss did not improve for 100 consecutive epochs. To improve training stability, biodimers training also employed gradient clipping with a maximum gradient norm of 5. Every training run included 10 warmup epochs, during which the learning rate was linearly increased from zero to the preset learning rate of $1 \times 10^{-4}$.

For the biodimers and NaCl cluster datasets, the targets were normalized by their standard deviation in the training set. The energy-to-forces loss weight ratio was set to 1:1 for biodimers and MgO surface datasets, while the NaCl cluster dataset used a ratio of 1:10.

For cumulene, PET was trained for $20\,100$ epochs with a batch size of 16, equal energy and forces weight, and a maximum learning rate of $2 \cdot 10^{-4}$. The learning rate was increased linearly from 0 over 100 epochs, and then reduced by 10% every 500 epochs over the entire training run. Architecturally, 5 attention layers and cosine cutoff function was employed to resolve the cumulenes. We found that increasing the number of attention layers, and adapting the cutoff function, were critical to resolve this benchmark task.

**CACE-LES** The CACE-LES model was originally presented in Cheng (a), with an additional modification introducing a long-range component in Cheng (b). Similar to MACE, CACE-LES is an equivariant message-passing neural network. After short-range message passing, invariant scalar pseudo-charges are predicted and passed to the long-range part of Ewald summation; the resulting energy contribution is added to the energy predictions of the short-range message-passing model. We use this model with the recommended setting of one message-passing step and, depending on the system, the following corresponding effective cutoff radii: $5.5 \, \text{Å}$ for the MgO surface, $5.29 \, \text{Å}$ for the NaCl cluster—these are the settings used in the original paper Kim et al.—and we chose a cutoff of $5 \, \text{Å}$ for biodimers and cumulene.

For the MgO surface and NaCl cluster datasets, models were taken from Kim et al.. For the cumulenes and biodimers datasets, the hyperparameters were as follows: 6 Bessel radial functions, $c = 8$, $l_{\max} = 3$, $\nu_{\max}$, $N_{\text{embedding}} = 2$, one message-passing layer, one-dimensional hidden variable, $\sigma = 1$, and $dl = 2$. Training followed the example in the original paper: the first 200 epochs used an energy loss weight of 0.1 and a forces loss weight of 1000, after which the energy loss weight was changed to 1, 10, and 1000 every 100 epochs, yielding a total of 500 epochs. The learning rate was $5 \times 10^{-3}$, with a step learning rate schedule decreasing it by a factor of 2 every 20 steps.

**4G-NN** The 4G-NN model was designed to tackle the benchmarks introduced by Ko et al., and introduced in that work. It consists of a shallow neural network acting on rotationally invariant features, predicting both a local energy contribution and an electronegativity, which is then used in a charge equilibration procedure that globally redistributes charges to minimize an energy expression. This process can be seen as a physics-inspired long-range message passing scheme iterated until a fixed point is reached. As the model requires charge labels to train, which are not available for all datasets, we did not train this model for our experiments but instead include results from Ko et al. Results for the 4G-NN model are only available for the NaCl cluster and MgO surface datasets. The cutoffs used in the original paper are as follows: $4.23\,\text{Å}$ for the MgO surface and $5.29\,\text{Å}$ for the NaCl cluster.

