# LEARNING LONG-RANGE REPRESENTATIONS WITH EQUIVARIANT MESSAGES: ADDITIONAL EXPERIMENTS FOR DISCUSSION PERIOD

To facilitate discussion, we report additional experiments. Following discussion, these results may be incorporated into the manuscript, but have not yet been included.

## 1 ABLATIONS OF LR BLOCK AND $l$

We repeated the experiments on the MgO surface, NaCl cluster, cumulene, and biodimers with LOREM models with $l = 0$ and $l = 1$, as well as without the long-range message-passing block. The parameter counts for no LR and $l = 0, 1, 2$ (in that order) are 839129, 1020300, 1020749, and 1021198.

Results can be seen in Table 1 and Figs. 1 to 3. Generally, higher $l$ improve error metrics. For experiments where only scalar long-range interactions are required (MgO surface, NaCl cluster), higher $l$ do not improve qualitative agreement. The cumulene example, which requires access to relative orientation between rotors, requires equivariant long-range interactions with $l = 2$ to be resolved. Removing the long-range message-passing block significantly increases error and renders the model unable to solve all the benchmark tasks.

Table 1: Root mean squared errors for energy $E$ and forces $\boldsymbol{F}$ for datasets used in Sec. 6.1 of the main text, for different LOREM variants. See Tab. 1 of the main text for details.

| Dataset | | LOREM No LR | LOREM LR $l = 0$ | LOREM LR $l = 1$ | LOREM LR $l = 2$ |
|---|---|---|---|---|---|
| MgO surface | $E$ (meV/at) | 2.234 | 0.063 | **0.062** | 0.064 |
| (Validation set) | $\boldsymbol{F}$ (meV/Å) | 61.487 | 5.870 | 5.284 | **4.076** |
| NaCl cluster | $E$ (meV/at) | 1.582 | 0.113 | **0.111** | 0.112 |
| (Validation set) | $\boldsymbol{F}$ (meV/Å) | 50.110 | 1.243 | **1.062** | 1.155 |
| Biodimers | $E$ (meV/at) | 8.259 | 0.302 | 0.329 | **0.222** |
| | $\boldsymbol{F}$ (meV/Å) | 16.452 | 1.677 | 1.725 | **1.646** |
| Cumulene | $E$ (meV/at) | 16.203 | 15.793 | 5.576 | **3.309** |
| | $\boldsymbol{F}$ (meV/Å) | 126.000 | 133.092 | 85.956 | **50.084** |

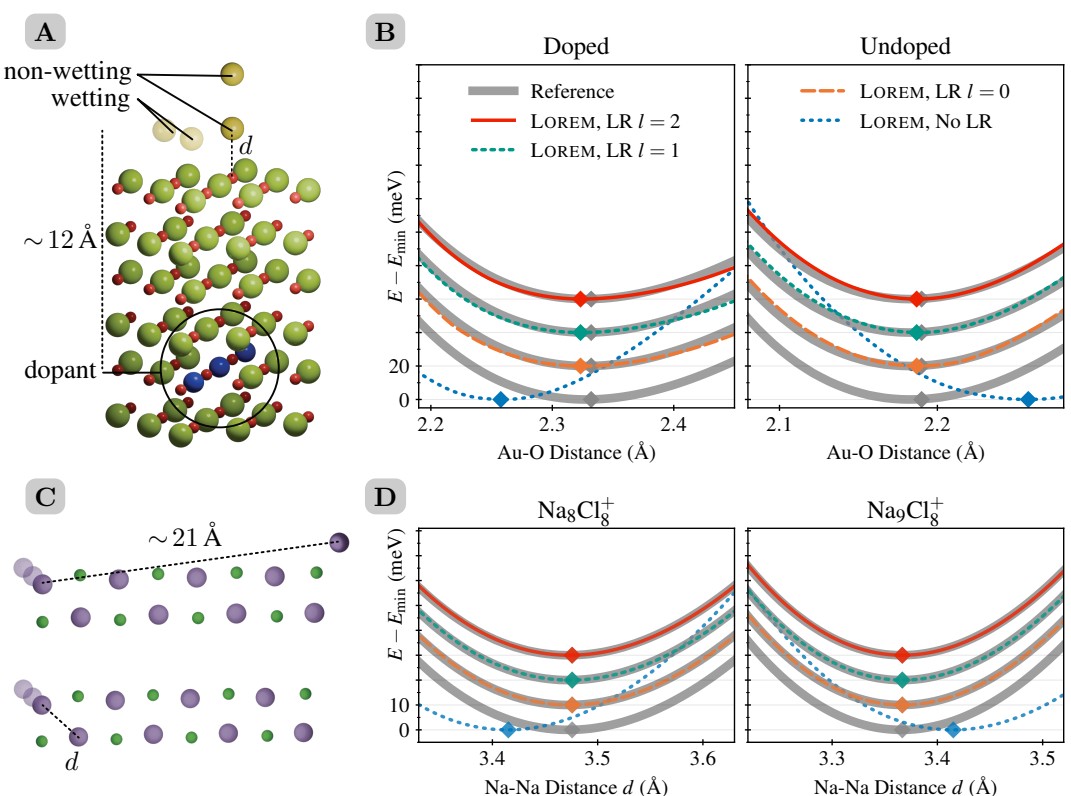

Figure 1: (**A**) Au$_2$ dimer on MgO surface, showing both wetting and non-wetting geometries, as well as the Al dopant. (**B**) Energy over distance $d$ for the non-wetting geometry for the doped and undoped surface. The minima are indicated with a diamond symbol; the reference energy curve is drawn in grey. Offsets are added to distinguish the curves and the value at the minimum is subtracted. (**C**) Na$_9$Cl$_8^+$ (top) and Na$_8$Cl$_8^+$ (bottom) cluster, the moving atom is marked with transparent copies of itself, and the distance of interest is labeled with $d$. (**D**) Energy over distance for both clusters.

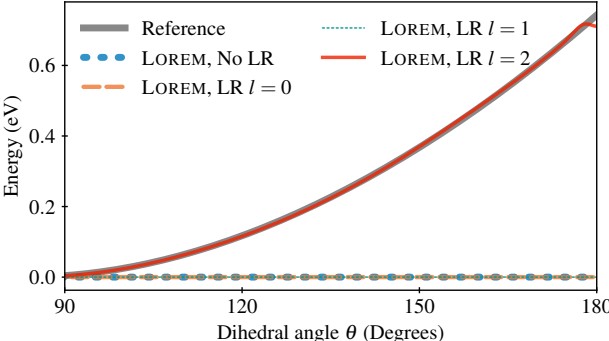

Figure 2: Energy profile over a $90°$ rotation of one rotor. The minimum value of each curve is subtracted before plotting.

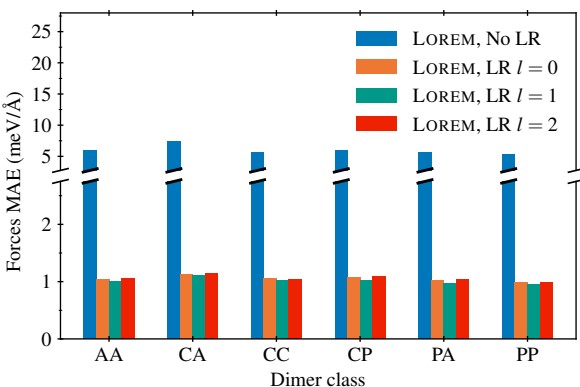

Figure 3: Mean absolute error on forces for different models on the different dimer classes: Apolar-apolar (AA), charge-apolar (CA), charge-charge (CC), charge-polar (CP), polar-apolar (PA), polar-polar (PP).

Table 2: Runtime of LOREM without (SR) and with long-range block (SR+LR), and their absolute and relative difference $\Delta$, reported for different crystal systems and numbers of atoms $N$.

| $N$ | Crystal | SR (ms) | SR+LR (ms) | $\Delta$ (ms) | $\Delta$ (%) |
|---|---|---|---|---|---|
| 16 | CsCl | 1.4 | 1.7 | 0.3 | 19.2 |
| 16 | NaCl primitive | 1.0 | 1.3 | 0.3 | 23.9 |
| 16 | zincblende | 0.9 | 1.2 | 0.3 | 25.4 |
| 32 | wurtzite | 3.9 | 4.3 | 0.4 | 9.4 |
| 48 | cu2o | 12.1 | 12.4 | 0.4 | 3.1 |
| 64 | NaCl cubic | 2.7 | 3.0 | 0.3 | 11.4 |
| 64 | wigner bcc | 4.2 | 4.6 | 0.4 | 8.5 |
| 64 | wigner fcc | 8.3 | 8.8 | 0.5 | 5.1 |
| 128 | CsCl | 10.1 | 10.4 | 0.4 | 3.4 |
| 128 | NaCl primitive | 4.8 | 5.1 | 0.3 | 6.5 |
| 128 | zincblende | 5.2 | 5.6 | 0.4 | 6.8 |
| 256 | wurtzite | 22.4 | 22.9 | 0.5 | 2.1 |
| 384 | cu2o | 103.8 | 104.4 | 0.6 | 0.5 |
| 512 | NaCl cubic | 18.2 | 18.8 | 0.5 | 2.9 |
| 512 | wigner bcc | 39.7 | 40.1 | 0.4 | 1.0 |
| 512 | wigner fcc | 89.8 | 90.6 | 0.8 | 0.9 |
| 1024 | CsCl | 71.1 | 72.1 | 1.0 | 1.3 |
| 1024 | NaCl primitive | 36.5 | 38.1 | 1.5 | 4.0 |
| 1024 | zincblende | 31.5 | 33.0 | 1.6 | 4.7 |
| 4096 | NaCl cubic | 161.0 | 170.6 | 9.7 | 5.7 |

## 2 RUNTIME BENCHMARK

To estimate the runtime of the LR block, we ran predictions (of energy and forces) for the benchmark structures used in (Loche et al.) for LOREM models with and without LR block. All benchmarks were repeated ten times and averaged, and executed on a NVIDIA H100 GPU. The results can be found in Table 2. For most system sizes and densities (or crystal structures), the LR block accounts for less than $10\,\%$ of overall runtime.

## REFERENCES

Philip Loche, Kevin K. Huguenin-Dumittan, Melika Honarmand, Qianjun Xu, Egor Rumiantsev, Wei Bin How, Marcel F. Langer, and Michele Ceriotti. Fast and flexible long-range models for atomistic machine learning. 162(14):142501. ISSN 0021-9606. doi: 10.1063/5.0251713.