# OpenReview forum: "Learning Long-Range Representations with Equivariant Messages"
_ICLR.cc/2026/Conference — ICLR 2026 Conference Withdrawn Submission_

### Official Review · Reviewer_JhS8 · 2025-10-23

**Soundness:** 1
**Presentation:** 1
**Contribution:** 1
**Rating:** 0
**Confidence:** 3

**Summary:**

This paper introduces a modification in equivariant MLIPs to handle long-range dependencies in periodic systems with O(N logN) complexity. The main change to existing equivariant MPNN architectures is to add a long-range dependencies block that considers all atoms in the cell, including those outside the usual cut-off radius. Results suggest that this new block is not always useful in practice, but according to the authors the proposed methodology avoids the careful tuning of hyper-parameters of equivariant architectures that do not have this block.

A disclaimer: I do not have a background in computational chemistry, but I have experience in ML and equivariant GNNs. Therefore, my evaluation will mostly focus on the quality of the work done from the machine learning perspective.

**Strengths:**

I appreciate the effort of the reviewers, especially in the introduction and background, in  giving readers a general understanding of the problem being solved. ICLR is a generalist machine learning conference, and as such one expects that readers are well aware of machine learning basics rather than knowledge of computational chemistry.

The paper is well written, the use of language appropriate and the organization are satisfactory. I believe that the results show, to some extent, that there is a positive trend in introducing the long-range module into equivariant architectures. The experimental setup is given ample room and results do not try to oversell the effectiveness of the approach. These aspects reflect the scientific integrity and rigor of the authors, something which we do not often see in the machine learning community and must be acknowledged. I am sure that the introduction of the long-range module can have a positive effect compared to the use of a few message passing layers, so my evaluation is not affected by the mixed results in the paper, which I instead find intriguing.

**Weaknesses:**

Overall, my assessment is that this paper is very accessible to a computational chemist well familiar with methods and techniques, but completely obscure to the average reader of the machine learning conference. The impact of this manuscript on a generalist conference is therefore low. This is reinforced by several aspects I will talk about later, for instance that the current implementation is no more scalable than others which have O(N^2) cost: being “amenable” to optimization does not mean that a differentiable implementation of techniques like PME is easy or even feasible today. Most papers on AI4Science I encountered tend to have two main issues, namely lack of clarity for an ML community due to the effort it takes to introduce terminology, operators etc., and unclear or incorrect ML evaluation procedures. The current version of the manuscript seems to suffer from both, but this does not mean it may be accepted as is to another venue less interested in ML aspects and more attentive on empirical results.

The manuscript clarity and self-containment for an ML conference can be improved especially starting from Section 2. Notions of tensors, orders of tensors, cell matrix, cut-off radiuses, are never formally introduced and all terminology is taken for granted. Again, this does not make the paper accessible to a ML conference. It is therefore unclear how to read Equation (1) as well as how it is evaluated in practice, given the summation over Z^3. While Appendix A gives an informal statement that the results are still equivariant, I would encourage the authors to include a full proof of it in the paper, which forces them to introduce all the appropriate terminology. The authors also assume that the readers/reviewers look at Appendix E, but the method should be self-contained in the main paper. Finally, the paragraph on “long-range message passing” is a good example of giving notions of spherical harmonics, spherical tensors etc for granted. It is unclear how potentials are actually split (line 232). I believe there is a lot of work to be done presentation-wise to make this paper readable by the average ML reader of ICLR. If the authors do not want to engage in such an effort, I would recommend submitting to another venue where specific domain knowledge is expected to be known.

Finally, I would like to mention that the abstract and introductions seem to over-sell a bit the impact of MLIPs based on first-principle data in the real world. I would agree that they are applicable today in material science contexts, but I do not really think they are “indispensable” today in biology, since a first-principle reference dataset on proteins is hard to find. I would suggest that the authors take more care when making such statements or better clarify why they believe this is really the case.

Methodologically speaking, the contribution seems a bit weak for an ML venue. The modification is rather trivial, which does not mean is bad, but the O(N^2) implementation makes me think that it does not add much to what excists already in the literature.

The empirical setup is problematic from a machine learning perspective. In Appendix F, the authors claim (line 960) that they tune the hyper-parameters of LOREM by hand on the validation set. First, this essentially means that on MgO surface and NaCl clusters the hyper-parameters are tuned on the “test” set. Indeed, if a 90/10 train-“validation” split is used, nothing prevents the authors from extracting a true validation set from the training data and using the “validation” set as the test set. This is a severe mistake because it biases the results of the first two datasets of Table 1. Second, while hyper-parameters may have been tuned for LOREM by hand, there is no mention to hyper-parameter tuning for the other baselines: the authors even argue  (line 319) that they used the “standard settings” of other baselines to carry out the comparison. Because of the no free lunch theorem, there exists no “standard setting” in machine learning, despite it being a common mistake. It may very well be the case that by tuning the number of MPNN layers, the cut-off radius etc., we might obtain comparable or even better performances with simpler equivariant MPNNs, regardless of the long-range module. Because no proper tuning of each method on each dataset has been done, the results are not reliable and cannot be considered “conclusive” to some extent (although I believe the authors that the new long-range block may have an statistical impact on results). This, in my opinion, is another severe methodological mistake. Finally, I do not understand why LODE was not included in the comparison, since it has been mentioned so early in the manuscript and seems very related to this work.

Concluding, as of now I believe there are grave mistakes in the empirical methodology and that the paper is not accessible for the ICLR community.

**Questions:**

Could the authors include LODE in the comparison and explain why the two approaches are different? From the introduction it appears both handle long-range dependencies in a similar way and compute equivariant features. What is the big difference?

---

> ### Author Response · Authors · 2025-11-22
>
> We thank the reviewer for their time and the detailed review. However, we do not understand the rating given the mentioned strengths of the work. To our understanding, a zero rating is reserved for fraudulent, incomplete, or egregiously wrong work. We do not believe that this is the case, and hope that our rebuttal makes this point.
>
> >  ... the current implementation is no more scalable than others which have O(N^2) cost: being “amenable” to optimization does not mean that a differentiable implementation of techniques like PME is easy or even feasible today
>
> This is not a theoretical consideration: The `torch-pme` and `jax-pme` packages ([Loche et al.](https://doi.org/10.1063/5.0251713)), cited and (the latter) used in the manuscript provide differentiable implementations of PME in both JAX and PyTorch. Additionally, linear-scaling methods like multi-level summation are also [available today](https://arxiv.org/abs/2510.05961). We therefore think it is correct to claim that the presented method scales $\mathcal{O}[N \log(N)]$ in periodic systems, and can, with multi-level summation, be rendered asymptotically linear scaling for both periodic and non-periodic systems. We have not yet experimented with multi-level summation, and suspect that the pre-factor may be prohibitive in practice; for this reason this is not discussed in detail in the manuscript.
>
> In the present work, we use plain Ewald summation for practical reasons: As seen in Fig. 4 of [Loche et al.](https://doi.org/10.1063/5.0251713), mentioned above, Ewald summation is faster for the system sizes (less than 1000 atoms) investigated here.
>
> > The manuscript clarity and self-containment for an ML conference can be improved especially starting from Section 2 ...
>
> While we understand the reviewer's frustration with AI4Science papers, we would like to point out that it is very difficult to both introduce all notation and domain expertise and fully discuss a new method and experiments. We have aimed to strike a balance, and would be happy to use the additional page for the camera-ready version to incorporate some suggestions from the reviwer.
>
> We would also like to point out that many AI4Science works have been published at ICLR (for instance [Gasteiger et al.](https://openreview.net/forum?id=B1eWbxStPH), [Liao and Smidt](https://openreview.net/forum?id=KwmPfARgOTD), [Gao et al.](https://openreview.net/forum?id=FhBT596F1X), [An et al.](https://openreview.net/forum?id=BBD6KXIGJL)), and GNNs applied to the atomistic domain are a well-established subfield. We therefore did not consider it neccessary to re-introduce all building blocks from scratch; we cite sources for these.
>
> > Finally, I would like to mention that the abstract and introductions seem to over-sell a bit the impact of MLIPs based on first-principle data in the real world. I would agree that they are applicable today in material science contexts, but ...
>
> While we agree that applications of MLIPs in biology are less common than in materials science and quantum chemistry, we are confident that MLIPs are a useful tool even today. For instance, recent works by [Unke et al.](https://doi.org/10.1126/sciadv.adn4397) and [Kabylda et al.](https://doi.org/10.1021/jacs.5c09558) demonstrate clearly that it is practically possible to perform MLIP simulations of protein fragments at QM accuracy. Nevertheless, we will be happy to weaken these statements.
>
> > The empirical setup is problematic from a machine learning perspective. In Appendix F ...
>
> We are aware that these experiments are not in line with typical ML practices and mention the non-standard use of a validation set repeatedly in the manuscript. We made this choice to be able to directly compare with existing published results, which use these splits as given and to our knowledge, are tuned in the same way.
>
> We would like to make three additional points:
>
> A. These datasets are mainly intended for *qualitative* benchmarks, which are extensively discussed in the remainder of the manuscript; these evaluations are performed on a split of the dataset that is not used for training or HP tuning. In any case, the quality of the match to reference is not the main factor in these experiments, the question is whether a given model can predict the correct behaviour *at all*.
>
> B. We did not extensively tune (training) HPs for these experiments -- in total, on the order of ten preliminary runs were executed before production. For this reason, the impact of HP tuning on the results in Table 1 was rather minor. The next comment begins with with the errors we would have reported if we had taken the "worst" model seen during HP tuning (excerpting runs that crashed/diverged; in the same units as in Table 1).
>
> We will continue our response in the next comment.

---

> ### Author Response · Authors · 2025-11-22
>
> | Dataset | Worst $E$ RMSE | Closest other $E$ RMSE | Worst $F$ RMSE | Closest other $F$ RMSE |
> | - | -- | -- | - | - |
> | MgO| 0.239| 0.071| 5.919| 5.971|
> | NaCl| 0.117| 0.210| 1.85| 9.784|
>
> The only change would therefore be a different ranking in $E$ for MgO -- a system where errors are already far below any practically relevant error bound. We are confident that the reported results are not a result of HP tuning.
>
> C. To verify that a standard HP tuning workflow would have produced similar results, we ran a HP sweep for an inner train/validation split (4000 samples/500 samples) with the following combinations of parameters: Initial LR $(0.001,0.0001)$, LR decay (linear, exponential), batch size $(32,64)$, energy/force loss weights $(0.5/0.5,100/1,1000/1)$ at $4000$ epochs of training. From the resulting models, we selected the ones with the best validation RMSE on energy and on forces. The results, again in the units of Tab. 1, are:
>
> For AuMgO task:
>
> | Model            | Test RMSE $E$ | Test RMSE $F$ |
> | - | - | - |
> | LOREM Manuscript | 0.064| 4.076|
> | Nearest other    | 0.071 (CACE)  | 5.971 (MACE)  |
> | Best $E$ model   | 0.064| 5.229|
> | Best $F$ model   | 0.078| 3.630|
>
> Here, one of the possible chosen models would have ranked second in energy (see B), and both would still be ranked first in forces.
>
> For NaCl task:
>
> | Model          | Test RMSE $E$ | Test RMSE $F$ |
> | - | - | - |
> | Paper          | 0.112| 1.155|
> | Nearest other  | 0.210 (CACE)| 9.784 (CACE)|
> | Best $E$ model | 0.076| 2.613|
> | Best $F$ model | 0.101| 1.473|
>
> Here, both alternative models are still ranked first. Indeed, even the worst LOREM model seen during the HP sweep performs better than the nearest other model.
>
> We also verified that both models perform similarly on the qualitative benchmarks of Fig. 2.
>
> > Second, while hyper-parameters may have been tuned for LOREM by hand, there is no mention to hyper-parameter tuning for the other baselines
>
> For CACE-LES and 4G-NN, we used published models for NaCl and AuMgO. We also trained CACE-LES ourselves on the biodimers and cumulene datasets using the provided scripts from this [GitHub repository](https://github.com/BingqingCheng/cace-lr-fit/tree/main), where the best model is selected from five runs with different energy and force weights in the loss. For the other models and datasets, a moderate amount of (training; see below) HP tuning was performed, roughly in line with LOREM.
>
> Most of the tuning for MACE and PET models was done on the cumulene dataset: for PET, we adjusted the number of attention layers and the cutoff function, and for MACE, we tuned max $l$. For the remaining datasets, we mainly focused on tuning hyperparameters that shape the training behavior - the number of epochs, the initial learning rate, and the learning-rate scheduler to improve convergence.
>
> We also would like to note that while we report errors in Tab. 1, our main results are reported in the remainder of the results section -- they center on the ability of different models to express physically reasonable long-range behaviour *at all*, and not solely the test set error.
>
> >  the authors even argue (line 319) that they used the “standard settings” of other baselines to carry out the comparison. Because of the no free lunch theorem, there exists no ...
>
> One of the points we make in this manuscript is exactly this: Standard GNNs **must** be tuned for specific systems, not to yield low errors, but to yield results that are not **fundamentally** incorrect. Some behaviour, such as the reaction in Fig. 5, cannot be learned by standard message passing with any (practical) set of hyper-parameters.
>
> > Because no proper tuning of each method on each dataset has been done, the results are not reliable
>
> As discussed above, we have (moderately) tuned HPs *except* the ones we are explicitly studying (cutoff, number of MP steps). We believe the results are conclusive to make the point they aim to make: Some physical behaviour requires long-range interactions to be included in the model architecture.
>
> >  I do not understand why LODE was not included in the comparison, since ...
>
> We mention LODE because it is a similar idea: Using potentials from Ewald summation (which is well understood and can be made efficient) to generate features with long-range information. However, LODE requires numerical integration, rendering it cumbersome to implement with AD systems. It also computes equivariant features of the electrostatic potential around an atom, rather than an equivariant potential from equivariant charges -- we do not expect it to have the same expressivity as our model, and did not believe it worthwhile to re-implement it. In [Huguenin-Dumittan et al.](https://doir10.1021/acs.jpclett.3c02375), the authors discuss that LODE is equivarlent to a multipole expansion of the *scalar* charge potential, and find that higher $l$ features contain limited information.

---

> > ### Comment · Reviewer_JhS8 · 2025-11-24
> >
> > I thank the authors for their response. To clarify, I could not find any indication in the ICLR's reviewer guide that a score of 0 corresponds to what the authors claim. Please point me to the specific sentence, in which case I am glad to raise the score to 2. My score reflects that this paper's presentation and empirical evaluation are not adequate for a generalist high-quality ML conference, as I extensively argued before. It is nothing personal nor it reflects any "frustration" with AI4Science papers. Let me re-iterate that this paper, like previous ones that have been published at these top conferences, which did *not* introduce in an accessible way (for the ML community) most necessary concepts, is not presented or executed in a way that, in my opininon, makes it interesting and valuable to the broader ICLR audience. As a reviewer, I strive to make the paper accessible and interesting to the specific audience I am reviewing for.
> >
> > As regard to some of the other points raised:
> >
> > - Since a differentiable implementation of PME exists (which I knew), I woud have expected the authors to use it to demonstrate that it provides similar performance, although slower for the considered small systems
> > - I cannot accept non-standard use of validation sets. If a procedure is wrong, then the correct way to go would be to properly re-tune all models according to robust ML practices (or some of them, specifying that the cost is high but at least the evaluation is sound). Again, I am quite sure that the results would not be far from what you obtain, but we are talking about maintaining high-quality standards of ML evaluations.
> > - The practice of tuning a model on one dataset and then fixing it on the other datasets is also incorrect.
> > - Based on the authors answer, LODE should have been included in the evaluation because it appears like a strongly related approach. The fact that it requires numerical differentiation is not a good argument to exclude the comparison.

---

> > > ### Author Response · Authors · 2025-11-28
> > >
> > > We thank the reviewer for their response. While we disagree on the suitability of the paper for ICLR, their comments have raised some issues that we have addressed above, and we appreciate their contribution to improving our work.
> > >
> > > > To clarify, I could not find any indication in the ICLR's reviewer guide that a score of 0 corresponds to what the authors claim. Please point me to the specific sentence, in which case I am glad to raise the score to 2.
> > >
> > > This was communicated to us by an AC during our own reviews of other papers, and we assumed that this was a general guideline. We will request clarification from the AC.
> > >
> > > > Since a differentiable implementation of PME exists (which I knew), I woud have expected the authors to use it to demonstrate that it provides similar performance, although slower for the considered small systems
> > >
> > > Benchmarks of Ewald summation vs PME were performed already in Loche et al., linked above, on similar hardware. Benchmarks of the SR part of the model were performed in response to other reviewers, see Tab. 2 of the revised supplementary material.
> > >
> > > > I cannot accept non-standard use of validation sets. [...]
> > >
> > > We have discussed this at length above and have nothing further to add. We will replace the current models with the models tuned on the inner training/validation split, and thank the reviewer for this suggestion, which will improve the manuscript.
> > >
> > > > The practice of tuning a model on one dataset and then fixing it on the other datasets is also incorrect.
> > >
> > > To our knowledge, this is standard practice in MLIPs, which often operate on similar datasets and are expected to perform well "out of the box". We have attempted to discuss our approach above and have nothing further to add.
> > >
> > > > Based on the authors answer, LODE should have been included in the evaluation because it appears like a strongly related approach. The fact that it requires numerical differentiation is not a good argument to exclude the comparison.
> > >
> > > LODE is not considered a standard or state-of-the-art method, and as-is is only suitable to generate features for linear and kernel regression approaches. We focus our experiments on end-to-end comparisons with other MLIPs. We have also discussed LODE's low equivariant information content, which is further discussed in the referenced work by [Huguenin-Dumittan et al.](https://pubs.acs.org/doi/full/10.1021/acs.jpclett.3c02375) We respectfully disagree that adding it to our comparison would improve the manuscript.

---

### Official Review · Reviewer_Gp2X · 2025-10-24

**Soundness:** 2
**Presentation:** 2
**Contribution:** 1
**Rating:** 2
**Confidence:** 4

**Summary:**

The authors introduce a new machine learning interatomic potential called LOREM that incorporates equivariant messages in long-range message passing. The long-range message passing is adapted from Ewald message passing. In their experiments, the authors find such long-range message passing to significantly improve PES modeling with small models especially when focusing on long-range effects.

**Strengths:**

* While the contribution is narrows down to replacing a scalar with a vector entity within the ewald summation, it is a well defined one and a worthwhile investigation.
* The paper is well written and can be understood easily.
* The proposed method performs well on the presented benchmarks.

**Weaknesses:**

The scope of the empirical evaluation is to limited and focuses on small datasets representing single PES where explicit long-range interactions take place:
* It would be great to see the performance on large-scale datasets such as OC20, Materials Project, etc.
* A proper ablation needs to take place as all methods differ in their short and long-range message passing. To ease comparisons, different short range models should be combined with different long-range models like Kosmala, Unke, Chmiela, etc.

**Questions:**

1. What is the runtime cost of the long range message passing compared to the short-range one?
2. How does LOREM compared to other long-range message passings like Ewald message passing (Kosmala et al.)
3. How does the parameter l affect the performance? Does equivariance add something or does standard Ewald message passing yield similar improvements?
4. How does the application to aperiodic systems works? A fundamental assumption in Ewald summation is that the system is periodic.

---

> ### Author Response · Authors · 2025-11-22
>
> We thank the reviewer for their review of our work.
>
> > It would be great to see the performance on large-scale datasets such as OC20, Materials Project, etc.
>
> For this submission, we focused on benchmarks that aim to isolate long-range behaviour. While performance on large-scale datasets would be interesting, it would be hard to isolate the impact of the proposed LR block versus other choices for datasets that are not specifically designed to emphasize long-range interactions.
>
> > A proper ablation needs to take place as all methods differ in their short and long-range message passing. To ease comparisons, different short range models should be combined with different long-range models like Kosmala, Unke, Chmiela, etc.
>
> Our approach in this work was to compare models end-to-end to give the most favourable conditions to all approaches.
>
> > What is the runtime cost of the long range message passing compared to the short-range one?
>
> We performed benchmarks to answer this question. We ran calculations on a single H100 for the benchmark structures from [Loche et al.](https://doi.org/10.1063/5.0251713), using the production LOREM model from the manuscript, toggling the LR block on and off. For moderate system sizes, the LR block accounts for approximately 10% or less of runtime. Note that overall runtime depends significantly on the density (and hence number of edges). Results can be found in Sec. 2 of the updated supplementary material.
>
> > How does LOREM compared to other long-range message passings like Ewald message passing (Kosmala et al.)
>
> One main difference compared to Ewald MP is that we fix the reciprocal space kernel to be $1/r^p$ where $p$ is a positive integer -- this allows LOREM to make use of standard methods from computational physics for the evaluation of such LR interactions. The numerical convergence in this case is also well studied. The other main difference is that we consider equivariance.
>
> > How does the parameter l affect the performance? Does equivariance add something or does standard Ewald message passing yield similar improvements?
>
> We have performed these ablations, and the results are presented in Sec. 1, Tab. 1, and Figs. 1–3 of the updated supplementary material. Higher $l$ generally improves error or at least does not degrade it; qualtitative performance on the benchmark tasks is only impacted if the problem requires it: The cumulene task can only be solved with $l\geq 2$.
>
> > How does the application to aperiodic systems works? A fundamental assumption in Ewald summation is that the system is periodic.
>
> Naiveley, the non-periodic implementation is just a quadratically scaling sum over all pairs.  However, methods with $\mathcal{O}[N]$ scaling in both periodic and aperiodic cases also exist and have recenly been implemented in the JAX machine learning framework ([Buchner et al.](https://arxiv.org/abs/2510.05961)). We have not experimented with this because for non-periodic systems of small and moderate size the naive sum is usually more efficient.

---

### Official Review · Reviewer_8Nn7 · 2025-10-29

**Soundness:** 1
**Presentation:** 3
**Contribution:** 2
**Rating:** 2
**Confidence:** 2

**Summary:**

The paper introduces a long-range message passing architecture for modeling non-local physical effects. The core idea is to build a symmetry-preserving architecture on top of Ewald summation where the global interaction is modeled by all-to-all interatomic interactions filtered by scalars inversely proportional to interatomic distances. The proposed model is evaluated on a range of scenarios where the extent of the dependence on the locality to the energy contribution varies a lot.

Overall, in the current form, the paper is not ready to be published because the paper would need a profound revision in regards to additional ablation studies and theoretical work about the equivariance for the core idea in the paper. The lack of these aspects leaves me very uncertain that many assertions regarding those aspects are fair and reasonable and is hard to neglect because they are tightly linked to the novelty and main contribution of the paper.

**Strengths:**

- The paper structure is organized well and adequately narrows down an issue of conventional MLIPs the authors address in the paper. The problem is timely and on point.
- The idea of incorporating Ewald summation sounds natural as a means to model long-range interactions.

**Weaknesses:**

**Vague description for the equivariance on the proposed representation:** Description about group equivariance/invariance is loose in general. When the core idea (1) is introduced, the type of group and the definition of its action are not clarified at all. Even when I assume the canonical orthogonal action (resp. translation) of $O(3)$ (resp. $\mathbb{Z}^{3}$) on $\mathbb{R}^{3}$, it is unclear how the equation (1) preserves those group actions, because the denominator has an additional term originated from the periodic boundary condition which I consider, at first glance, is supposed to be fixed throughout the group actions. Hence, it is unclear if all the empirical results are reflecting any influence stemmed from the equivariance of the architecture (1) the authors envisioned. Besides, while the authors point to Appendix A for the detail, this section is not helpful since the description remains on a high level and it is hard to access whether the experiment mentioned in this section is fair or not.

**Evaluation of computational cost:** The relative additional computational cost, incurred by the equivariant long-range architecture, against the overall computational complexity is not clarified, and it is hard to access the significance of the potential overall cost reduction, when using an advanced method such as the particle mesh Ewald summation as discussed in Appendix. Since it seems that the experiments do not use those advanced algorithms, I believe it is crucial to detail the computational cost at least from a theoretical point of view.

**Ablation study:** The ablation study looks largely missing. The unit cell $C$ is introduced as a hyperparameter, but its choice is not clarified in the large portion of the experiments and the impact to the performance is also not evaluated systematically. Another missing study is the ablation of an inverse law of interatomic potential functions. One (loose) view of the proposed equivariant long-range representation (1) is "equivariant" GraphTransformer filtered by a scalar function inversely proportional to the interatomic distance. While the experiments show an advantage of the proposed model, it is still unclear which of these additional components (i.e., the global aggregation and the physics-inspired inverse distance function) significantly contributed to the performance gain in the experiments.

**Questions:**

- What group is assumed in the equation (1)? What space does the group act on?
- How is the equation (1) proven to be equivariant to the group action?
- How does the choice of the unit cell impact on the performance?
- What if the inverse law is omitted from the equivariant architecture?

---

> ### Author Response · Authors · 2025-11-22
>
> We thank the reviewer for their review.
>
> > When the core idea (1) is introduced, the type of group and the definition of its action are not clarified at all.
>
> The group is named in line 97 of the manuscript. In principle, we deal with both, $SO(3)$ and $O(3)$, the group of rotations (and inversions), applied to Euclidean vectors in 3D space. The current implementation focuses on $SO(3)$, but the method can be trivially extended to include inversions. We'll be happy to make this clearer in a revised manuscript.
>
> > Even when I assume the canonical orthogonal action [...] it is unclear how the equation (1) preserves those group actions, because the denominator has an additional term originated from the periodic boundary condition which I consider, [...] to be fixed [...]
>
> We believe that there may be a misunderstanding here, potentially due to the rather abridged notation used in Eq. 1. We consider rotations $g\in G$ (where $G=SO(3),O(3)$) acting **jointly** on all the geometric inputs of the MLIP: The positions $R_i \in \mathbb{R}^3$ and the three cell vectors $C_a \in \mathbb{R}^3$. The cell vectors are commonly concatenated into a $3 \times 3$ cell matrix. The sum over $\mathbb{Z}^3$ signifies an (infinite) sum over all possible periodic replicas of the positions: For each $R_i$, we sum over $R_i + n_1 C_1 + n_2 C_2 + n_3 C_3$ where each $n_a$ ranges over all integers. We combined these offsets into a vector $n$ in Eq. 1.
>
> Overall, the denominator in Eq. 1 is therefore of the form $| V |^p$ where $V \in \mathbb{R}^3$ is a **vector** and $|\cdot |$ signifies the vector norm. The denominator is therefore **invariant** under rotations. The enumerator is equivariant by construction, as we work with equivariant features. Equivariance of the overall expression follows from the linearity of the action of rotations: A sum of equivariants is equivariant.
>
> We will clarify the notation in future revisions.
>
> > Hence, it is unclear if all the empirical results are reflecting any influence stemmed from the equivariance of the architecture (1) the authors envisioned.
>
> Our claim is not that the results are purely due to equivariance, we simply assert our method does not disrupt the existing equivariance of the model. Equivariant neural networks are well established for MLIPs. There is an ongoing lively discussion about their merits versus "unconstrained" models, which we consider an interesting avenue for future work.
>
> > Appendix A [...] is not helpful [...] and it is hard to access whether the experiment mentioned in this section is fair or not.
>
> We will revise this section to more explicitly discuss the argument made above. What concerns does the reviewer have regarding the "fairness" of this experiment?
>
> > The relative additional computational cost, incurred by the equivariant long-range architecture, against the overall computational complexity is not clarified, [...] I believe it is crucial to detail the computational cost at least from a theoretical point of view.
>
> We have designed our method such that it can be implemented with existing algorithms for the fast evaluation of electrostatic (or other inverse power law) potentials. Using particle-mesh Ewald summation (PME), the asymptotic cost in periodic systems is $\mathcal{O}[N \log(N)]$. Naive evaluation scales $\mathcal{O}[N^2]$ in non-periodic systems. However, methods with $\mathcal{O}[N]$ scaling in both cases also exist and have recently been implemented in the JAX machine learning framework ([Buchner et al.](https://arxiv.org/abs/2510.05961)).
>
> We agree that a practical benchmark of the additional computational cost of the LR message passing block would be useful, and have run this experiment. The results can be found in Tab. 2 of the updated supplementary material. For moderate system sizes, the LR block accounts for approximately 10% or less of runtime. Overall runtime depends significantly on the density (and hence number of edges).
>
>
> We will continue our response in the next comment.

---

> > ### Author Response · Authors · 2025-11-22
> >
> > >  The unit cell $C$ is introduced as a hyperparameter, [...]
> >
> > The unit cell is an **input** of the model, defining how the atoms are tiled in 3D space. It is not a hyperparameter and cannot be tuned. We will clarify this in future revisions of the manuscript.
> >
> > > Another missing study is the ablation of an inverse law of interatomic potential functions.
> >
> > As the main objective method of our method is its compatibility with existing algorithms from computational physics, we did not perform ablations of the functional form of the potential: Existing algorithms *require* an inverse power law to function. As mentioned in the manuscript, we briefly investigated different exponents, but did not find a positive impact on results.
> >
> > > While the experiments show an advantage of the proposed model, it is still unclear which of these additional components (i.e., the global aggregation and the physics-inspired inverse distance function) significantly contributed to the performance gain in the experiments.
> >
> > We have performed ablations varying $l=0,1,2$ and disabling the LR block altogether; the results can be found in Tab. 1 and Figs. 1–3 of the updated supplementary material.
> >
> > We did not investigate global aggregations without a distance-dependent filter, as we consider these approaches outside the design space of the model, which is intended to use inverse power-law potentials. Speculatively, we predict that global aggregation of equivariant features can resolve the cumulene, NaCl, and MgO benchmarks, which require no distance information to be conveyed over long distances, but it cannot improve performance on biodimers or the SN2 reactions.
> >
> > > What group is assumed in the equation (1)? What space does the group act on?
> >
> > Either $SO(3)$ or $O(3)$, acting either on the inputs (vectors in $\mathbb{R}^3$) or equivariant features (vectors in $\mathbb{R}^{2l+1}$).
> >
> > > How is the equation (1) proven to be equivariant to the group action?
> >
> > As discussed above, the proof is trivial and relies only on two facts: (a) Scalars are rotationally invariant, and (b) rotations act as linear maps on vectors in $\mathbb{R}^{2l+1}$. Eq. 1 is simply a sum over vectors in $\mathbb{R}^{2l+1}$ and equivariance follows from linearity; all other objects are invariants.
> >
> > > How does the choice of the unit cell impact on the performance?
> >
> > As discussed above, it is not a free choice. Generally speaking, the unit cell impacts the runtime of the model because it determines the number of reciprocal-space vectors that must be considered in Ewald summation.
> >
> > > What if the inverse law is omitted from the equivariant architecture?
> >
> > The use of an inverse power law is required to use Ewald summation and related methods, and we therefore consider it a fixed constraint, rather than a free choice to be ablated. We therefore did not perform experiments without it.
> >
> > In general, if the inverse power law is dropped and replaced by a general, learned, function, we recover equivariant, global, message passing.

---

> ### Comment · Reviewer_8Nn7 · 2025-11-26
>
> I thank the authors for their response. Some of the concerns were addressed. I have some follow-up questions and comments regarding the architecture's equivariance.
>
> How exactly is the unit cell chosen? Since its choice is not free, I speculate that there should be a specific method for deriving cells from given atomic systems. I am asking this question because the proof of the equivariance is NOT trivial, even with the authors' clarification. This is because an equivariant function needs to be able to produce group-compatible outputs without assuming access to group elements applied to the input, and for now it is unclear whether it also holds for the method the authors used to generate the cells, given the explanation. Therefore, it is still impossible to ensure that the overall architecture is equivariant to the authors' aforementioned group action and that the equivariant experiment in Appendix A is conducted with the assumption.

---

> > ### Author Response · Authors · 2025-11-26
> >
> > We thank the reviewer for their timely response and are happy to hear that we've been able to address their concerns.
> >
> > > How exactly is the unit cell chosen? Since its choice is not free, I speculate that there should be a specific method for deriving cells from given atomic systems.
> >
> > The cell vectors are not chosen by us, they are part of the inputs processed by the model. Together with the positions (and the atomic numbers), they define the arrangement of atoms in space for which LOREM (and other MLIPs) makes a prediction:
> >
> > In condensed matter physics, a solid (or liquid) is described as an infinite set of position vectors in $\mathbb{R}^3$, relative to some coordinate origin. Rotations are considered as rotations of *all* of these vectors at once, or equivalently, as the rotation of all three basis vectors of $\mathbb{R}^3$.
> >
> > To make this infinite set of positions possible to model computationally, the standard approximation of condensed matter physics is made: Instead of treating each position as independent, they are reduced to a periodic repetition of some *finite* number $N$ of positions. This description consists of two parts: A set of  positions $R_i \in \mathbb{R}^3$ ($i=1...N$), and three cell vectors $C_a \in \mathbb{R}^3$ ($a=1,2,3$). The infinite "bulk" system is then described as including, for each $i$, $R_i + n_1 C_1 + n_2 C_2 + n_3 C_3$ where each $n_a$ ranges over all integers.
> >
> > By convention, the $N$ positions are chosen to be enclosed in the parallelepiped spanned by the cell vectors. This combined object is called the "unit cell" and the infinite system is constructed by seamlessly tiling it in space. A rotation of the infinite set of positions with a rotation matrix $g$ reduces to multiplying $g$ with all $R_i$ and $C_a$. This rotates all positions at once.
> >
> > For any given periodic arrangement of atoms, there are many different choices of unit cell, all describing the same overall set of points in $\mathbb{R}^3$. Our model, and to our knowledge all MLIPs acting on periodic systems, is invariant to this choice: For any choice of unit cell (with the same number of atoms) describing the same periodic set of positions in space, it will return the same energy and forces. This is because MLIPs act effectively on the periodic system and not the positions and cell vector separately. For this reason we also do not consider separate transformations of positions and cell.
> >
> > To summarise, positions and cell vectors, together with the atomic number of each atom, define the inputs of our model -- they are not chosen or generated by us. In the periodic datasets in the manuscript, we used the positions and cells as given.

---

### Official Review · Reviewer_C9Nx · 2025-10-31

**Soundness:** 2
**Presentation:** 2
**Contribution:** 3
**Rating:** 6
**Confidence:** 3

**Summary:**

The paper presents an architecture based on SO(3) equivariant spherical harmonic features to build a machine learning potential.

The architecture uses message passing (MP) over the spherical tensor and scalar

The architecture also contains a long-range component that takes the output from the MP and computes an Ewald summation over the nodes of a tensor, which represent the charges.

The author tested on $5$ datasets built to test if ML model can represent charges.

The authors compare with various state-of-the-art models: MACE, PET, CACE-LES, and 4G-NN.

The introduction, background, and related work are well described.

In section 4, the authors use the definition of indices and subscripts introduced in the background, which makes the reading of this session hard.

The authors claim that equation (1) is equivariant. They also refer to Appendix A for further demonstration and refer to the numerical confirmation.

Unfortunately, Equation (1) is not equivariant, and the session in the Appendix does not really prove anything if not repeating that Eq.a is equivariant.

In general, the authors refer to the Ewald summation and the efficient implementation. This is used when simulating a periodic system, but the paper only provides results with point evaluation. This is also recognized by the authors in lines 163-164.

Equation (1) is equivariant either with an isolated system (n==0) or when the periodic boundary is rotated with the system, which is not assumed by the authors.

Further, equation (1) is referred to as implementing message passing. Even if the authors provide two references (Kosmala, Grisafi) claiming the same, equation (1) only corresponds to the aggregation of a message passing procedure; therefore, the name (of the session and of the module) contains two descriptors that do not really apply in this case.

In the sentence 164-165 ("The method..."), it is not clear what is meant by "accommodates".

In sentence 214-215, could you give more information on 1) "expanded in Bernstein polynomials multiplied with  fcut, the cosine cutoff function, yielding ρij" 2) how "Orientations rij are expanded in spherical harmonics  Yl  ij."

In line 216 "k" is not defined.

In line 226, why M P_i are called scalar features?

In line 227 "likewise" seems not applicable, since elements are transformed differently.

In line 227, self-tensor product is not defined.

In line 232: how are potential splited?

In line 234: "set of different exponents" could you clarify? what did you try? not clear what it this sentence means.

One main contribution of the paper is to consider tensorial Q. It would be nice to see what happens with the scalar version only and understand why the "self-tensor product" of ${}^M S_i^l$ is necessary.

**Strengths:**

The evaluation of the method shows a good improvement over previous models.

Overall, the evaluation is well presented and complete.

The paper is generally clear, but is lacking in the method description (section 4)

In general, the paper addresses an important aspect of modeling long-range interactions.

**Weaknesses:**

Equation (1) (and the all session) is, in theory, the main contribution of the paper, but it has many inaccuracies.

**Questions:**

Please check my previous comments.

---

> ### Author Response · Authors · 2025-11-22
>
> We thank the reviewer for their time and in-depth review.
>
> > Equation (1) is equivariant either with an isolated system ($n==0$) or when the periodic boundary is rotated with the system, which is not assumed by the authors.
>
> This appears to be a misunderstanding shared by reviewer 8Nn729: We consider rotations of **all** inputs, i.e., positions and cell vectors (in the periodic case). The reviewer is correct that this is a requirement for the equivariance of Eq. 1. To our knowledge, there is no physical reason to treat rotations of the positions and the cell separately, and we did not consider this case.
>
> We will clarify this in a future revision.
>
> > In general, the authors refer to the Ewald summation and the efficient implementation. This is used when simulating a periodic system, but the paper only provides results with point evaluation.
>
> Indeed, Ewald summation applies to periodic systems. In the non-periodic case, the naive fallback is the $\mathcal{O}[N^2]$ quadratic summation of all pairs. More efficient algorithms, such as multi-level summation (see other comments) exist. For small system sizes, we expect the simple all-to-all method to be more efficient.
>
> In the manuscript, both periodic and non-periodic systems are considered: AuMgO and Biodimers are periodic cases, the rest aperiodic.
>
> > Further, equation (1) is referred to as implementing message passing. [...] the name (of the session and of the module) contains two descriptors that do not really apply in this case.
>
> To our knowledge, message passing ([Gilmer et al.](https://proceedings.mlr.press/v70/gilmer17a.html)) consists of
>
> - A message function $M(e_{ij}, h_i, h_j)$ of node features $h_{i}$ and $h_j$, as well as edge features $e_{ij}$
> - An aggregation into a message per node $m_i = \sum_j M_{ij}$
> - An update of node features based on $m_i$ and $h_i$
>
> Eq. 1 describes a particular (very simple) form for the message function $M$, while Ewald summation implements the sum to obtain $m_i$. The update function is described in Fig. 1 and Appendix E.
>
> We hope this clarifies why we call our method long-range message passing.
>
> > In the sentence 164-165 ("The method..."), it is not clear what is meant by "accommodates".
>
> We agree that this may be unclear, we meant to indicate that the LR features decay asymptotically with distance.
>
> >In sentence 214-215, could you give more information on 1) "expanded in Bernstein polynomials multiplied with fcut, the cosine cutoff function, yielding ρij" 2) how "Orientations rij are expanded in spherical harmonics Yl ij."
>
> We will add an explanation to the appendix.
>
> > In line 216 "k" is not defined.
>
> Indeed. $k$ enumerates message passing steps.
>
> > In line 226, why M P_i are called scalar features?
>
> LOREM processes both equivariant features ($S^l$), which transform under rotations, and invariant (scalar) features ($P$) which do not. Since operations on equivariant features are both more expensive and more restricted, the feature dimensions are different and they are processed separately.
>
> > In line 227, self-tensor product is not defined.
>
> We considered this to be a standard operation, but agree that it may be preferable to have a self-contained explanation, which we will add.
>
> > In line 232: how are potential splited?
>
> To perform Ewald summation in parallel, we concatenate the scalar charge $q$ with the different equivariant charges: For $l=1$, there are three components $Q_{m=-1,0,+1}^1$ and for $l=2$ five, with $m=-2,-1,0,+1,+2$. All these features are flattened into an array of length $10$, yielding $10$ potentials at each atom (note that we also include the $l=0$ equivariant component, which is technically redundant with the scalar charge, but we keep it for simplicity). The concatenation is then performed in reverse, i.e., the first entry yields the scalar potential, the second the another $l=0$ scalar potential, the next three the $l=1$ potential components, and so on. We will explain this in the manuscript.
>
> > In line 234: "set of different exponents" could you clarify? what did you try?
>
> We tried $p=1,2,3$, i.e., adding additional exponents, but did not find an improvement on the biodimers dataset.
>
> We will improve this in the revised version.
>
> > One main contribution of the paper is to consider tensorial Q. It would be nice to see what happens with the scalar version only
>
> We ran this experiment, together with $l=1$ and without the LR part; the results are presented in Tab. 1 and Figs. 1-3 of the updated supplementary material. While performance is similar on most benchmark tasks, $l\leq 1$ fails to capture the interactions in cumulene. This is expected: Cumulene is the only case that cannot be solved by invariant long-range interactions, as it requires knowledge of the relative orientation of the rotors. (While distances should in principle be sufficient, the differences are very small and seem to be insufficient for both LOREM and CACE-LES.) Higher $l$ also generally yield lower test set errors.

---

### Note · Authors · 2025-12-02

**Comment:**

We would like to withdraw our submission "Learning Long-Range Representations with Equivariant Messages" (ID: 17288) from consideration for ICLR 2026. We sincerely thank the reviewers and our area chair for the time and effort they have dedicated to our submission. After careful internal discussion, we have concluded that we are unwilling to proceed under the current review conditions: We attempted to engage constructively with the reviewers through detailed rebuttals, but -- exacerbated by the shortened discussion period -- were unable to conclude a substantive, or constructive, discussion with any of them.

**Withdrawal Confirmation:**

I have read and agree with the venue's withdrawal policy on behalf of myself and my co-authors.